# Aberrant Transferrin and Ferritin Upregulation Elicits Iron Accumulation and Oxidative Inflammaging Causing Ferroptosis and Undermines Estradiol Biosynthesis in Aging Rat Ovaries by Upregulating NF-Κb-Activated Inducible Nitric Oxide Synthase: First Demonstration of an Intricate Mechanism

**DOI:** 10.3390/ijms232012689

**Published:** 2022-10-21

**Authors:** Stephen Cho Wing Sze, Liang Zhang, Shiqing Zhang, Kaili Lin, Tzi Bun Ng, Man Ling Ng, Kai-Fai Lee, Jenny Ka Wing Lam, Zhang Zhang, Ken Kin Lam Yung

**Affiliations:** 1Department of Biology, Faculty of Science, Hong Kong Baptist University, Kowloon, Hong Kong SAR 999077, China; 2Golden Meditech Center for NeuroRegeneration Sciences, Hong Kong Baptist University, Kowloon, Hong Kong SAR 999077, China; 3School of Chinese Medicine, LKS Faculty of Medicine, The University of Hong Kong, Pokfulam, Hong Kong SAR 999077, China; 4JNU-HKUST Joint Laboratory for Neuroscience and Innovative Drug Research, College of Pharmacy, Jinan University, Guangzhou 999077, China; 5School of Public Health, Guangzhou Medical University, Guangzhou 999077, China; 6School of Life Sciences, Faculty of Science, The Chinese University of Hong Kong, Shatin, N.T., Hong Kong SAR 999077, China; 7Department of Obstetrics and Gynaecology, LKS Faculty of Medicine, HKU, Pokfulam, Hong Kong SAR 999077, China; 8Department of Pharmaceutics, UCL School of Pharmacy, University College London, 29-39 Brunswick Square, London WC1N 1AX, UK; 9Department of Pharmacology & Pharmacy, LKS Faculty of Medicine, HKU, Pokfulam, Hong Kong SAR 999077, China

**Keywords:** transferrin, ferritin, NF-κB/iNOS, ovarian oxi-inflammaging, ferroptosis

## Abstract

We report herein a novel mechanism, unraveled by proteomics and validated by in vitro and in vivo studies, of the aberrant aging-associated upregulation of ovarian transferrin and ferritin in rat ovaries. The ovarian mass and serum estradiol titer plummeted while the ovarian labile ferrous iron and total iron levels escalated with age in rats. Oxidative stress markers, such as nitrite/nitrate, 3-nitrotyrosine, and 4-hydroxy-2-nonenal, accumulated in the aging ovaries due to an aberrant upregulation of the ovarian transferrin, ferritin light/heavy chains, and iron regulatory protein 2(IRP2)-mediated transferrin receptor 1 (TfR1). Ferritin inhibited estradiol biosynthesis in ovarian granulosa cells in vitro via the upregulation of a nuclear factor kappa-light-chain-enhancer of activated B cells (NF-κB) and p65/p50-induced oxidative and inflammatory factor inducible nitric oxide synthase (iNOS). An in vivo study demonstrated how the age-associated activation of NF-κB induced the upregulation of iNOS and the tumor necrosis factor α (TNFα). The downregulation of the keap1-mediated nuclear factor erythroid 2-related factor 2 (Nrf2), that induced a decrease in glutathione peroxidase 4 (GPX4), was observed. The aberrant transferrin and ferritin upregulation triggered an iron accumulation via the upregulation of an IRP2-induced TfR1. This culminates in NF-κB-iNOS-mediated ovarian oxi-inflamm-aging and serum estradiol decrement in naturally aging rats. The iron accumulation and the effect on ferroptosis-related proteins including the GPX4, TfR1, Nrf2, Keap1, and ferritin heavy chain, as in testicular ferroptosis, indicated the triggering of ferroptosis. In young rats, an intraovarian injection of an adenovirus, which expressed iron regulatory proteins, upregulated the ovarian NF-κB/iNOS and downregulated the GPX4. These novel findings have contributed to a prompt translational research on the ovarian aging-associated iron metabolism and aging-associated ovarian diseases.

## 1. Introduction

The ovaries age much faster than extragonadal organs, as well as the testes [1]. The aging of different organs shows many differences from one another. The ovarian function lasts from menarche until the cessation of menstruation. The hormonal function of the testis starts in utero, shows interruptions between neonatal life and puberty, and resumes with spermatogenesis, with only a small decline, until an advanced age. Andropause in the male progresses only slowly [2]. The aging processes of the ovary and testis are distinctively different [3,4].

Normal ovarian aging involves a follicle and oocyte deterioration, culminating in estrogen deficiency and the onset of menopause [5]. The only treatment option currently available to alleviate menopausal symptoms and prevent its related diseases is hormone replacement therapy (HRT), but there are safety concerns, such as the risks of cancer and stroke [6]. A better understanding of the mechanism of ovarian aging is crucial to finding alternative and safer treatments. Aging-associated systemic iron (Fe) accumulation due to an elevated serum level of ferritin (Ft, the major iron storage protein) has been demonstrated in menopausal women [7,8] and in various aged mouse tissues, such as the ovaries [9]. However, the consequence and mechanisms of aged-related ovarian Fe accumulation are still obscure.

The hepatic peptide hormone hepcidin interacts with its receptor ferroportin, which is a cellular iron exporter, and mediates Fe absorption and mobilization. Hepcidin binding to ferroportin brings about the ubiquitination, endocytosis, and proteolysis of ferroportin. Hepcidin is subject to feedback regulation by the plasma Fe concentration and Fe stores and is negatively regulated by the activity of the dominant iron consumers. When the level of storage Fe is elevated, hepcidin is produced, which feeds back to the gastrointestinal tract and to the placenta in pregnant women, to inhibit the further absorption of exogenous Fe in order to prevent an Fe overload. Hepcidin suppresses Fe release from the reticuloendothelial system to the circulating transferrin (Tf). In reticuloendothelial blockade, storage Fe is not released to the circulating Tf, resulting in a high serum Ft and a low Tf saturation level. In hemochromatosis, which is a genetic defect of hepcidin activity, the normal feedback suppression of intestinal Fe absorption is not present with the consequence of an Fe overload. The genetic or inflammatory overproduction of hepcidin ensues in Fe deficiency. Hepcidin and ferroportin also play a role in host defense and inflammation. Hepcidin synthesis is induced by inflammatory interleukin-6 and activin B [10,11].

Here, we used 2D gel-based proteomics as a rapid screening method to explore the target proteins in the ovaries of naturally aging SD rats, to examine changes in expression of the aforementioned proteins, and to uncover the key processes in normal ovarian aging with a dysregulated Fe metabolism. In this study, we hypothesized that an aberrant Tf and Ft upregulation in the naturally aging rat ovary induces an Fe accumulation and then oxi-inflamm-aging. A deficiency of Nrf2/antioxidant cytoprotection induced via the NF-κB-mediated inducible nitric oxide synthase (iNOS) results in oxi-inflamm-aging. Specifically, we aimed to investigate (1) whether two molecules involved in Fe regulation by negative feedback, Tf (via transferrin receptor) and Ft, are aberrantly upregulated as aging progresses; (2) the aging-associated changes in the Fe levels and oxidative stress; (3) if estradiol (E2) levels are depressed in Ft-treated primary rat ovarian granulosa cells via the NF-κB-induced iNOS pathways; and (4) if NF-κB-induced oxidative stress (OS) and inflammation increases with aging and we further confirmed these findings using a single intraovarian injection of an adenovirus which was expressing Ft and Tf in young rats, respectively. This study could contribute to an understanding of the theory of Fe homeostasis disruption-mediated inflammation in ovarian aging, which will add to the current knowledge and prompt related translational and clinical investigations, and provide tools for menopause prediction, leading to treatments for premature ovarian aging, and prolong the fertility period through the management and manipulation of early folliculogenesis. 

## 2. Results

### 2.1. Age-Associated Changes in the Expression of Proteins in Rat Ovarian Tissue Demonstrated by Two-Dimensional Gel-Based Proteomics

The changes in the proteins of the ovaries of SD rats aged 3, 9, 12, 16, and 22 months, corresponding to early reproductive maturity, the active reproductive period, the late reproductive period, the menopause/post-reproductive period, and the senescence period, respectively, were examined. A proteomics analysis disclosed that the expression of eleven proteins was altered by more than 3-fold (Figure 1 and Table 1). Nine proteins (ferritin heavy chain Fth1, Phb, heat shock protein family A (Hsp70) member 5 Hspa5, lactate dehydrogenase B chain Ldhb, fatty acid binding protein 3 Fabp3, ferritin light chain Ftl1, hemoglobin subunit alpha 1Hba1, transferrin Tf, and selenium binding protein 2 Selenbp2) were upregulated with Ftl1 enhanced to the greatest extent (by 5.33-fold), whereas two proteins (glutathione S-transferase theta-3 Gstt3 and carbonyl reductase 1 Cbr1) were downregulated (Figure 1 and Table 1). A Blast2GO analysis was performed for the annotation, mapping, and InterPro analysis of the identified proteins (Appendix A). A GO analysis was performed at levels 2 to 7 for the three main categories (cellular components, molecular functions, and biological processes) with cutoffs of five, five, and three for protein distribution, respectively (Appendix A). A molecular function analysis revealed that six proteins (Fth1, Ldhb, Gstt3, Cbr1, Ftl1, Hba1) exhibited oxidoreductase activity and five proteins (Fth1, Hspa5, Ftl1,Hba1, Tf) displayed metal ion binding properties (level 5 molecular function), of which four (Fth1, Ftl1, Hba1, and Tf) could bind iron ions (level 7 molecular function) (Appendix A). This implies that a dysregulated Fe metabolism may play a key role in ovarian aging. Additionally, GO annotation revealed that Tf upregulated I-κB kinase/NF-κB signaling also regulated tumor necrosis factor (TNF) production and the Phb protein response to nitric oxide (NO) (Figure 2, Figure 3 and Figure 4, Appendix A). These results imply that the aberrant upregulated Fth1, Ftl1, and Tf induced the dysregulated Fe metabolism and upregulated the I-κB kinase/NF-κB signaling intermediates, which were dysregulated by the metabolic processing of nitrogen compounds. 

**Figure 1 ijms-23-12689-f001:**
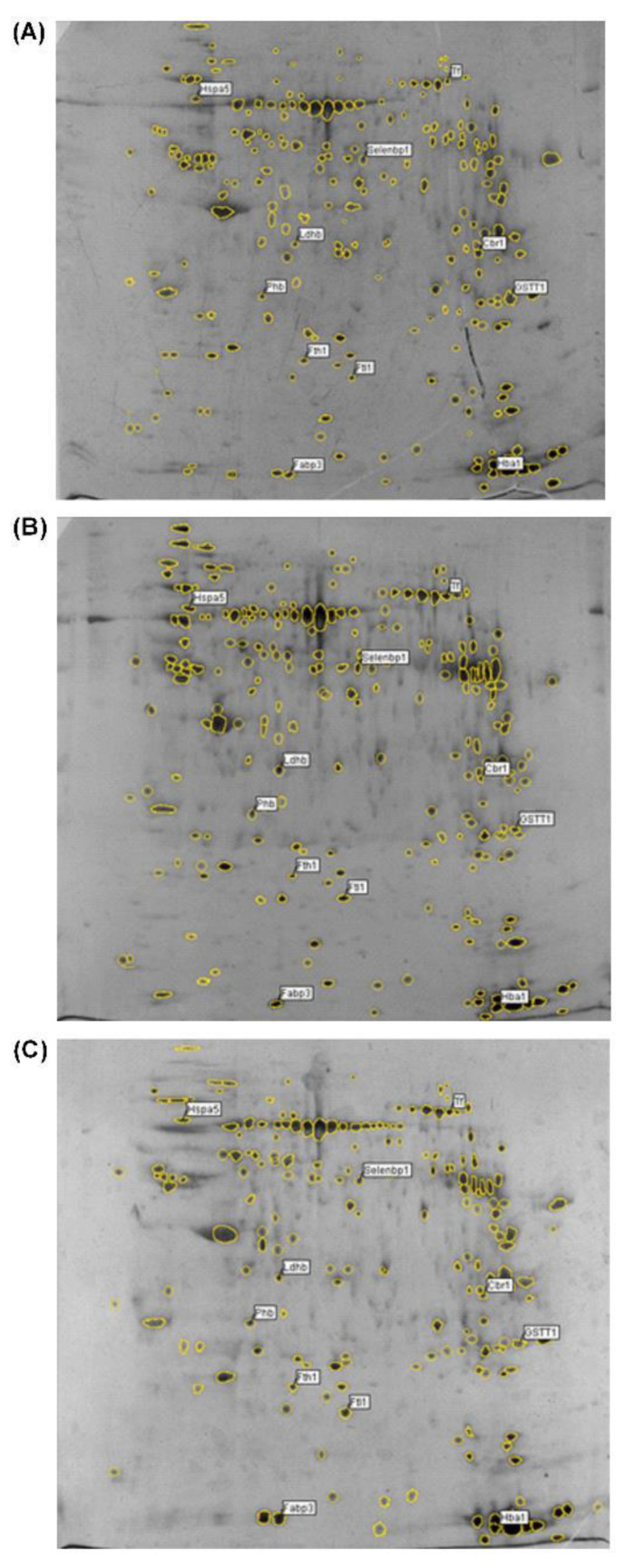
**Proteome profiles of ovarian tissues of naturally aging rats (3, 16, and 22 months of age) by two-dimensional gel electrophoresis.** (**A**) 3-month-old, (**B**) 16-month-old, and (**C**) 22-month-old. Analysis was performed using 120 μg of protein loaded on 17 cm non-linear ReadyStripTM IPG strips (pH 3–10), followed by separation on 12% SDS-PAGE gel and silver staining. Molecular weight is from high (top) to low (down) and pH is from low (left) to high (right). Identified protein spots on the gel images are labeled with their abbreviated names. Full names and UniProtKB accession numbers are shown in Table 1.

### 2.2. Aberrant Upregulation of Transferrin, Ferritin, Transferrin Receptor 1, and Iron Regulatory Protein 2 in Ovaries of Naturally Aging Rats

Intracellular Fe homeostasis is mainly coordinated by the regulation of TfR1 mRNA stabilization and Ft mRNA translation (post-translational regulation) by the IRP/IRE system [12]. To validate and extend the proteomics data in aging ovaries, Western blotting was used to quantify changes in iron import proteins (Tf and TfR1), export protein (ferroportin), storage proteins (FTH/Ftl chain), and iron-regulatory proteins (IRP1/2) in the ovaries of naturally aging rats (3, 9, 12, 16, and 22 months of age). The levels of Tf, FTH, Ftl chains, TfR1 and IRP2 were upregulated whereas the levels of ferroportin and IRP1 showed slight and statistically insignificant fluctuations (Figure 2). Specifically, Tf was significantly upregulated in the ovaries of 12-, 16-, and 22-month-old rats compared with those of 3-month-old rats and 9-month-old rats (Figure 2A,B). The ferroportin level was slightly upregulated in the ovaries of 9-month-old rats compared with those of 3- and 12-month-old rats, and slightly upregulated in the ovaries of 12-, 16-, and 22-month-old rats (Figure 2A,C). The Ftl (Figure 2A,D), FTH (Figure 2A,E), TfR 1 level (Figure 2F,G), and IRP 2 level (Figure 2F,I) were significantly upregulated in the ovaries of 12-, 16-, and 22-month-old rats compared with 3-month-old rats and 9-month-old rats. These results firstly indicate that the ovarian Tf, Ft, and TfR 1 were all aberrantly upregulated, leading to Fe accumulation in the ovaries of naturally aging rats. The upregulation in the level of the IRP2 in the absence of changes in the level of the IRP1 does not necessarily imply the regulation of the TfR1 mRNA by the IRP2. In this case, the transregulatory activity of the IRP1 is of central importance. The present investigation reveals a rise in the level of nitrosative stress, to which the IRP1 is very sensitive, owing to its iron–sulfur cluster [13]. Thus, the IRP1 binding activity and not the increase in the IRP2 level may be associated with the increase in the TfR1 mRNA.

**Figure 2 ijms-23-12689-f002:**
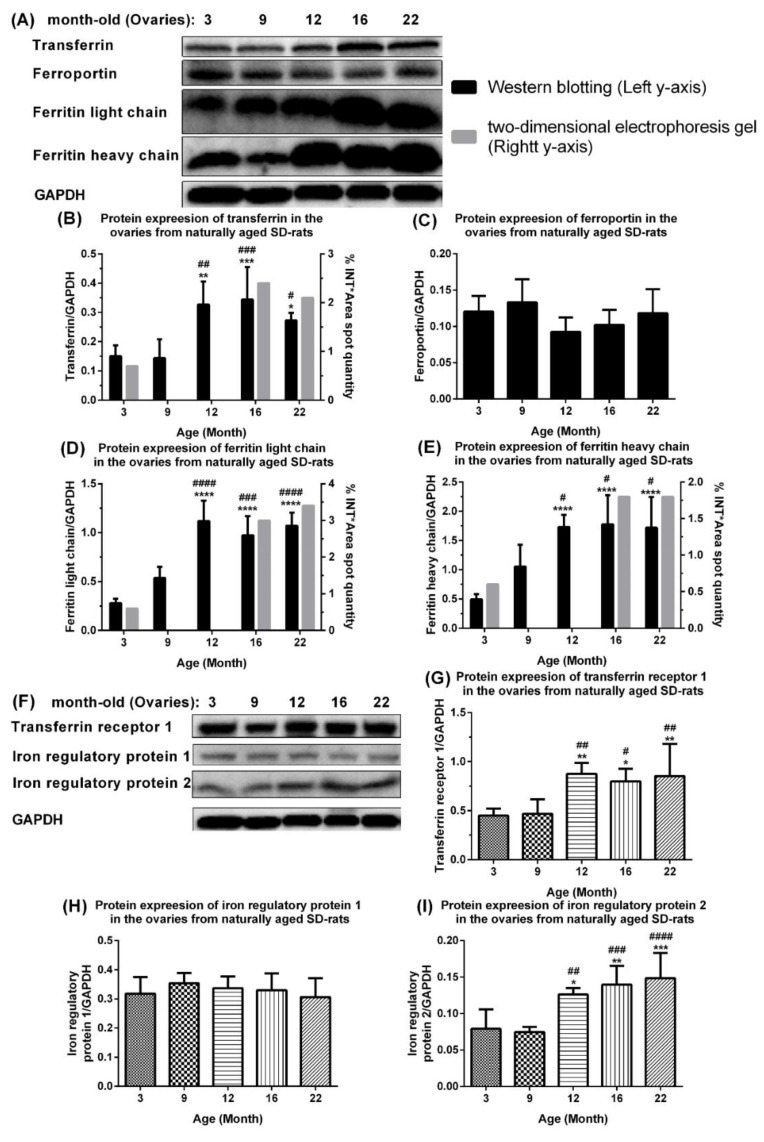
**Potential mechanism of age-associated iron accumulation in ovaries of naturally aging rats.** (**A**) Western blots and quantification of (**B**) transferrin, (**C**) ferroportin, (**D**) ferritin light chain, and (**E**) ferritin heavy chain in ovaries of naturally aging SD rats. (**F**) Western blots and quantification of (**G**) transferrin receptor 1, (**H**) iron regulatory protein 1, and (**I**) iron regulatory protein 2 in ovaries of naturally aging SD rats. Data are presented as mean ± SD (*n* = 6). The data between different age groups were compared using one-way ANOVA followed by Tukey’s multiple comparison test. *, *p* < 0.05; **, *p* < 0.01; ***, *p* < 0.001; ****, *p* < 0.0001 compared with 3-month-old rats. #, *p* < 0.05; ##, *p* < 0.01; ###, *p* < 0.001; ####, *p* < 0.0001 compared with 9-month-old rats.

Changes in the serum estradiol level, ovarian index, ovarian levels of labile ferrous iron, total iron, total nitrate/nitrite, 3-nitrotyrosine (3-NT), and 4-hydroxy-2-nonenal (HNE) occur during the process of natural aging in rats.

Iron is essential for oxygen transport, DNA biosynthesis, and energy production [14,15,16]. Ferritin is an iron-storage protein. Serum ferritin is a marker of the total body iron store. Its level in females shows fluctuations during menstruation and pregnancy and increases with age, while the total iron-binding capacity decreases [17], which leads to an iron accumulation due to the lack of active iron removal [12,18]. Exogenous/endogenous estrogen has been used to relieve diseases/disorders in extragonadal tissues caused by excess iron [19]. To correlate changes in the iron metabolism with age, we examined the changes in the aging physiological endpoints, including the ovarian indexes (ovarian weight over body weight × 100%), serum estradiol level, ovarian iron level, and ovarian oxidative stress. The ovarian weights and ovarian index were significantly lower in 12-, 16-, and 22-month-old rats compared with those of 3-month-old rats (Figure 3A), with 3- and 9-month-old rats (Figure 3B), respectively. Linear regression revealed that the serum estradiol level showed a significant decline with age (*p* = 0.0041, slope = −0.8405 ± 0.1055, *r*^2^ = 0.9549; Y = −0.8552X + 24.28), e.g., 22-month-old rats compared with 3-month-old rats (Figure 3C). The ovarian levels of the labile ferrous iron levels (Figure 3D), total labile iron (ferrous and ferric) level (Figure 3E), and the oxidation markers total nitrate/nitrite, 3-NT and HNE as well as IRP2 were upregulated in the ovaries of older rats, e.g., 12-, 16-, and 22-month-old rats compared with those of younger rats, e.g., 3- and 9-month-old rats. IRP1 did not show striking changes (Figure 3E–J).

**Figure 3 ijms-23-12689-f003:**
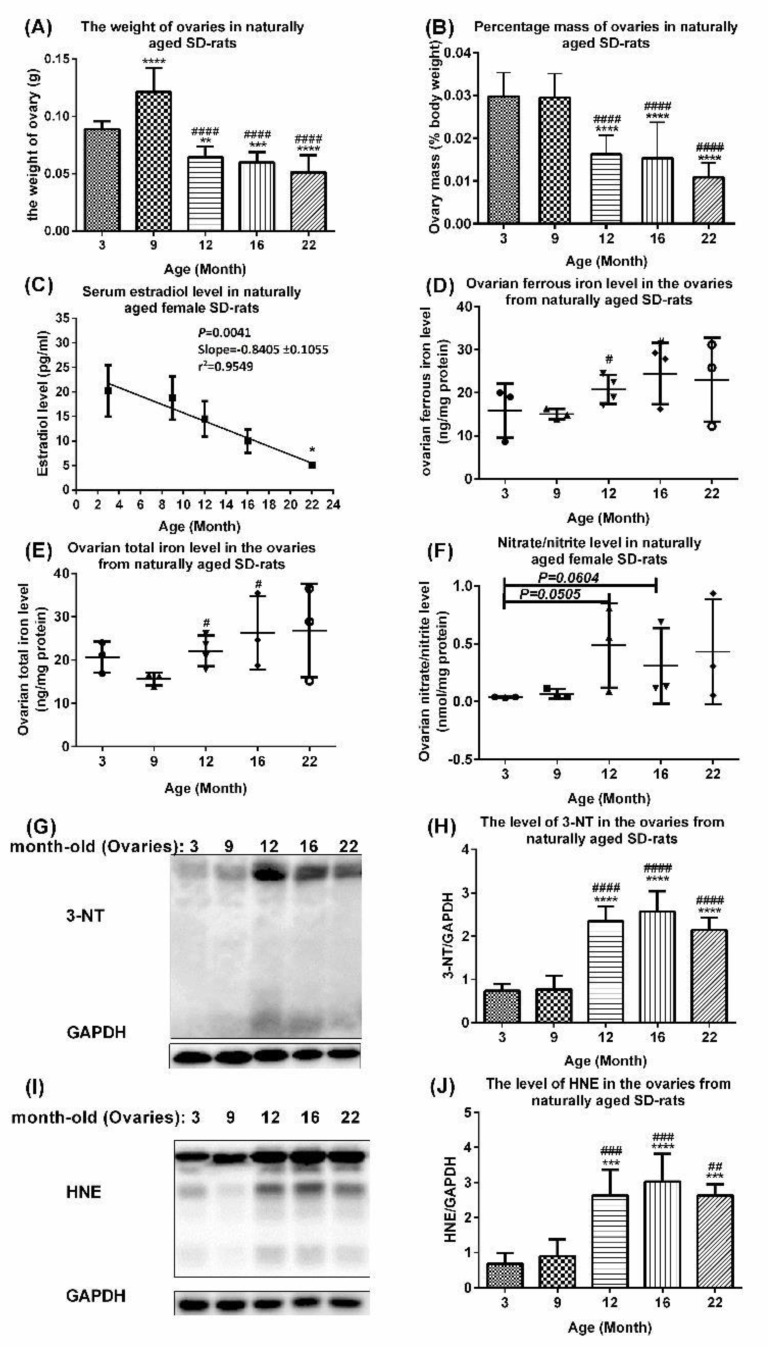
**Age-associated changes in serum estradiol level and other physiological changes in ovaries of aging rats.** (**A**,**B**) Changes in (**A**) ovarian weight and (**B**) percentage ovarian weight (as % of body mass) in aging rats. Data are presented as mean ± SD (*n* = 6). (**C**) Linear regression of serum estradiol levels in naturally aging SD rats at different ages (*n* = 5–6). (**D**,**E**) Changes in levels of ovarian labile (**D**) ferrous iron and (**E**) total iron (ferrous/ferric) in aging rats. Data are presented as mean ± SD (*n* = 3–4). (**F**–**J**) Age-associated increase in oxidative stress and (**F**) nitrate/nitrite level in aging rat ovaries. Data are presented as mean ± SD (*n* = 3). The differences between various age groups were compared using unpaired *t*-test. (**G**–**J**) Protein damage and lipid oxidation, (**G**,**H**) 3-NT levels, and (**I**,**J**) HNE levels in ovaries of naturally aging SD rats. Data are presented as mean ± SD (*n* = 6). The differences between various age groups were compared using one-way ANOVA followed by Tukey’s multiple comparison test in (**A**,**B**) and (**G**–**J**), and compared using unpaired *t*-test in (**C**), (**D**,**E**), and (**F**). In (**A**,**B**), ** *p* < 0.01; ***, *p* < 0.001 and ****, *p* < 0.001 compared with 3-month-old rats; ####, *p* < 0.0001 compared with 9-month-old rats. In (**C**), *, *p* < 0.05 compared with 3-month-old rats. In (**D**,**E**), #, *p* < 0.05 compared with -month-old rats. In (**G**–**J**) ***, *p* < 0.001; ****, *p* < 0.0001 compared with 3-month-old rats; ##, *p* < 0.01; ###, *p* < 0.001, compared with 9-month-old rats; ####, *p* < 0.0001, compared with 16-month-old rats.

Ferritin induces the downregulation of estradiol secretion via the upregulation of the NF-κB/iNOS pathway in isolated rat granulosa cells. Excess iron elicits iron-dependent inflammation and oxidative stress (OS), resulting in tissue damage and functional disorders. Cellular chronic inflammation and reactive oxygen species (ROS)-mediated cellular death are the fundamental features of aging [20], but their interrelationship in the ovarian granulosa cells has not been fully elucidated. Ovarian granulosa cells can take up ferritin [21]. In this study, treatment with iron-containing ferritin (20, 60, and 180 μg) for 12 h downregulated the estradiol in primary ovarian granulosa cells (Figure 4A) but upregulated the NF-κB p65 (Figure 4B,C), NF-κB p50 (Figure 4B,D), and iNOS (Figure 4B,E) levels in the culture medium of the primary ovarian granulosa cells in a dose-dependent manner. The upregulation was due to an increased transcription, increased mRNA stabilization, and/or increased protein stabilization. 

**Figure 4 ijms-23-12689-f004:**
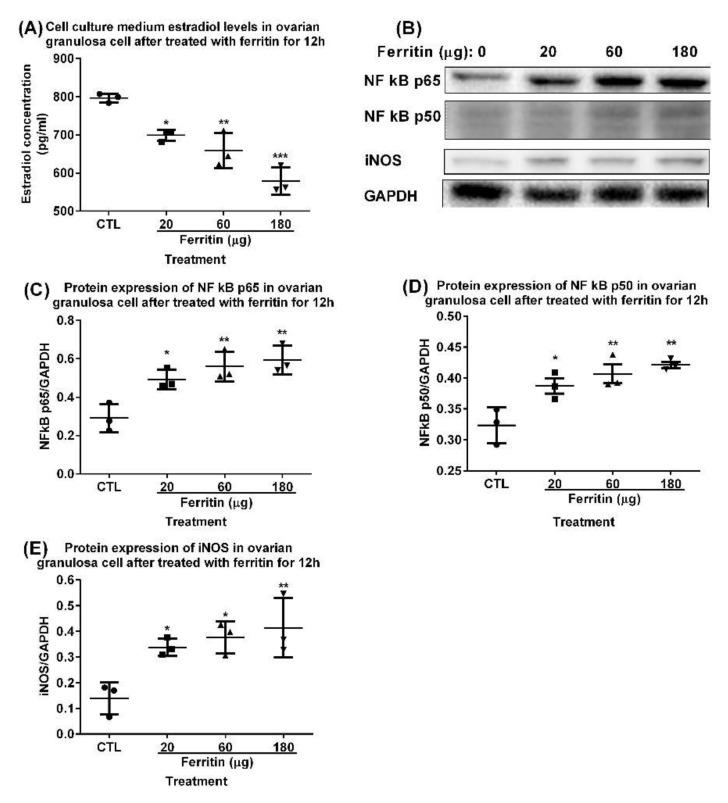
**Effects of ferritin on rat granulosa cells in primary culture.** (**A**) Estradiol levels in the culture medium of ovarian granulosa cells treated with ferritin (20, 60, and 180 μg) (iron-containing) for 12 h. (**B**) Western blots and quantification of (**C**) NF-κB p65, (**D**) NF-κB p50, and (**E**) iNOS in ovarian granulosa cells. Data are presented as mean ± SD (*n* = 3). The differences between groups were compared using one-way ANOVA followed by Tukey’s multiple comparison test. *, *p* < 0.05; **, *p* < 0.01; ***, *p* < 0.001 compared with the control group.

### 2.3. Age-Associated Aberrant Activation of NF-Κb/Inos and TNF Alpha Pathways in Ovaries of Aging Rats

The production of peripheral blood mononuclear cell cytokines including the TNFα significantly increases with age in healthy menopausal women [22]. Transcriptional factor NF-κB is the most important upstream oxidant-sensitive inflammatory signaling factor of the TNFα [23,24]. Our results confirmed that NF-κB-induced oxidative stress and inflammation increased with age in the ovaries of aging rats. The protein expression levels of the NF-κB p65 (Figure 5A,B) and NF-κB p50 (Figure 5A,C) were significantly upregulated in the ovaries of 12-, 16-, and 22-month-old rats compared with 3-month-old rats and 9-month-old rats. The levels of A20 showed a trend of downregulation with age in ovaries of 3- to 22-month-old rats (Figure 5A,D), although levels in 9-, 12-, and 22-month-old rats were slightly downregulated compared with those of 3- and 16-month-old rats. The levels of I-κB alpha showed a trend of upregulation in the ovaries of 12-, 16-, and 22-month-old rats compared with those of 3- and 9-month-old rats, although this was not statistically significant (Figure 5A,E). The upregulation was due to an increased transcription, increased mRNA stabilization, and/or increased protein stabilization. The levels of pI-κB alpha were upregulated in the ovaries of 12-, 16-, and 22-month-old rats compared with those of 3- and 9-month-old rats, with levels in 12-month-old rats significantly higher than those in 3-, 9-, and 22-month-old rats. Interestingly, the NF-κB binding activities were significantly higher in the ovaries of 12- and 16-month-old rats compared with those of 3-month-old rats (*p* < 0.05) and 9-month-old rats (*p* < 0.01) (Figure 5F,G), whereas the levels in 22-month-old rats were upregulated compared with those of 3- and 9-month-old rats, but the levels were downregulated compared with 12- and 16-month-old rats (Figure 5F,G). The protein expression levels of iNOS (Figure 5H,I), TNF α (Figure 5H,J), and TNFR1 (Figure 5H,K) were significantly upregulated in ovaries of 12-, 16-, and 22-month-old rats compared with those of 3-month-old rats and 9-month-old rats.

**Figure 5 ijms-23-12689-f005:**
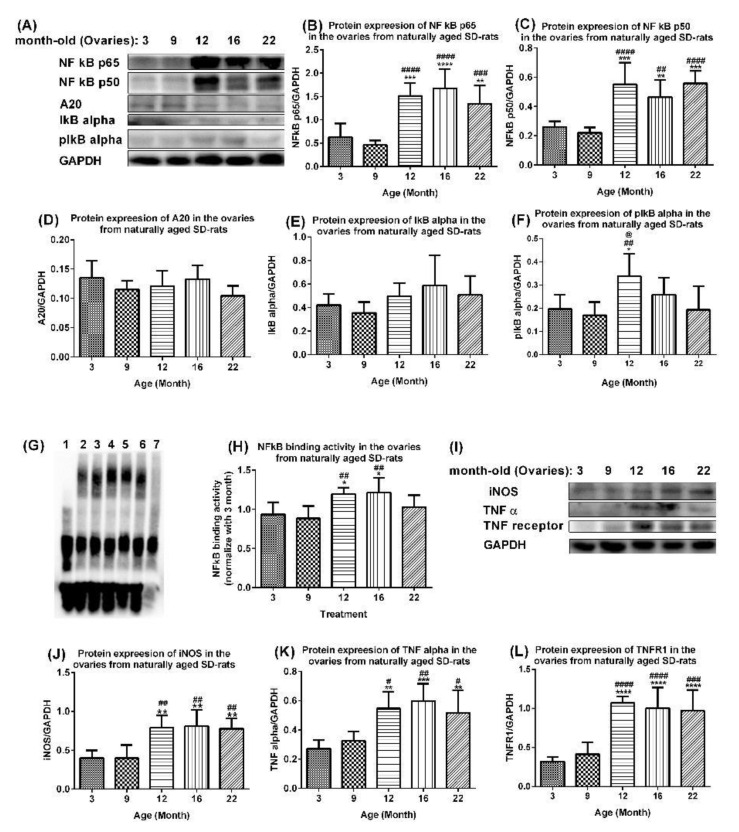
**Activated NF-κB-mediated upregulation of oxidative and inflammatory factors in ovaries of aging rats.** (**A**) Western blots and quantification of (**B**) NF-κB p65, (**C**) NF-κB p50, (**D**) A20, (**E**) I-κB alpha, and (**F**) pI-κB alpha, in ovaries of naturally aging SD rats. Data are presented as mean ± SD (*n* = 6). (**G**,**H**) NF-κB binding activity in ovaries of naturally aging SD rats. Data presented as mean ± SD (*n* = 6). Lane 1, blank control; Lane 2, 3 M (3-month-old); Lane 3, 9 M; Lane 4, 12 M; Lane 5, 16 M; Lane 6, 22 M; Lane 7, competitive control: 9 M and 200-fold molar excess of unlabeled DNA. (**I**) Western blots and quantification of (**J**) iNOS, (**K**) TNF α, and (**L**) TNFR1 in ovaries of aging rats. Data are presented as mean ± SD (*n* = 6). The differences between various age groups were compared using one-way ANOVA followed by Tukey’s multiple comparison test. *, *p* < 0.05; **, *p* < 0.01; ***, *p* < 0.001; ****, *p* < 0.0001 compared with 3-month-old rats. #, *p* < 0.05; ##, *p* < 0.01; ###, *p* < 0.001; ####, *p* < 0.0001 compared with 9-month-old rats. @, *p* < 0.05; compared with 22-month-old rats.

### 2.4. Age-Associated Aberrant Inhibition of Nrf2/GPX4 Pathway in Ovaries of Aging Rats

The activated Nrf2/Keap1 antioxidant signaling and overexpression of antioxidant enzymes [25] attenuated an NF-κB-mediated inflammation in vitro. Meanwhile, the Nrf2/Keap1 signaling pathway plays a central role in the antioxidation system by regulating the expression of antioxidant enzymes such as GPX4, GPX1, and GSTP1. We firstly reported this pathway in the normal ovarian aging process in SD rats in vivo. The protein expression levels of Nrf2 were upregulated in the ovaries of 9-month-old rats compared with those of 3-month-old rats (*p* < 0.05), but were significantly downregulated in those of 12-, 16-, and 22-month-old rats compared with those of 9-month-old rats (Figure 6A,B). The protein expression levels of Keap1 generally increased with age in the ovaries of 12-, 16-, and 22-month-old rats compared with those of 3- and 9-month-old rats, and was most significant in 16-month-old rats compared with 9-month-old rats (*p* < 0.05) (Figure 6A,C). The protein expression levels of GPX4 were upregulated in the ovaries of 9-month-old rats, but downregulated in the ovaries of 12-, 16-, and 22-month-old rats compared with 3-month-old rats and 9-month-old rats (Figure 6A,D). The protein expression levels of GPX1 showed a slightly downregulated trend in the ovaries of 3-, 9-, and 12-month-old rats that changed to a slightly upregulated trend in those of 16- and 22-month-old rats (Figure 6A,E). The levels of GSTP1 were slightly upregulated with age in the ovaries of 3-, 9-, 12-, 16-, and 22-month-old rats (Figure 6A,F). Compared with those of 9-month-old rats, the levels of SOD1 were slightly upregulated in the ovaries of 3-month-old rats, but downregulated in those of 12-, 16-, and particularly 22-month-old rats (Figure 6A,G). The protein expression levels of SOD2 were upregulated with age and were significantly higher in the ovaries of 16- and 22-month-old rats compared 3- and 9-month-old rats (*p* < 0.01) (Figure 6A,H). The levels of CAT were slightly downregulated with age in the ovaries of 3-, 9-, 12-, 16-, and 22-month-old rats (Figure 6A,I).

**Figure 6 ijms-23-12689-f006:**
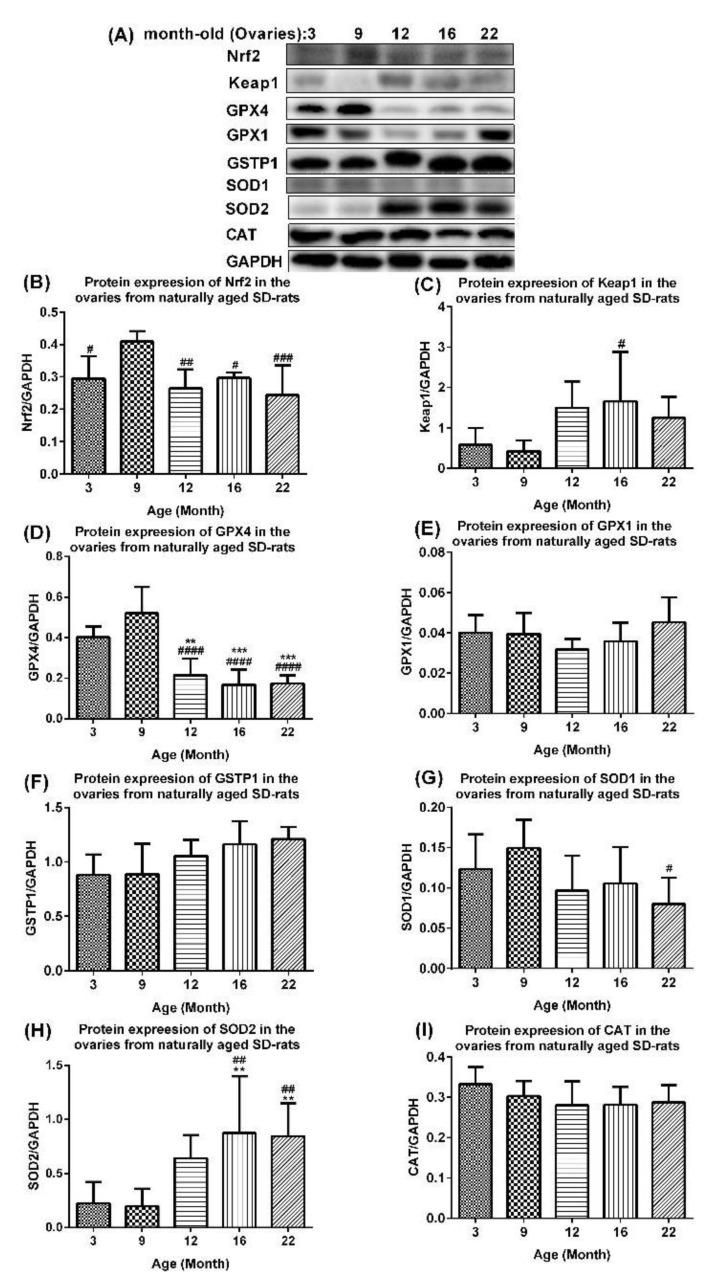
**Potential mechanism of mitigated antioxidant defense in ovaries of naturally aging rats.** (**A**) Western blots and quantification of (**B**) Nrf2, (**C**) Keap1, (**D**) GPX4, (**E**) GPX1, (**F**) GSTP1, (**G**) SOD1, (**H**) SOD2, and (**I**) CAT in ovaries of aging rats. Data are presented as mean ± SD (*n* = 6). The differences between age groups were compared using one-way ANOVA followed by Tukey’s multiple comparison test. **, *p* < 0.01; ***, *p* < 0.001, compared with 3-month-old rats. #, *p* < 0.05; ##, *p* < 0.01; ###, *p* < 0.001; ####, *p* < 0.0001, compared with 9-month-old rats.

### 2.5. Intraovarian Injection of Adenovirus Overexpressing Ferritin Light Chain/Heavy Chain and Transferrin Induced Upregulation of NF Kb P65/P50 Mediated Inos and Downregulation of GPX4 in Ovaries of 3-Month-Old Rats

To study the direct effect of ferritin and transferrin on the ovarian microenviroment, we investigated the effects of a bilateral intraovarian injection of Ad-ferritin light chain (Ad-Flc), Ad-ferritin heavy chain (Ad-Fhc), and Ad-transferrin (Ad-Tf), respectively, on serum E2 and the ovaries of 3-month-old female SD rats, including the protein expression levels of NF-κB p65/p50, iNOS, and GPX4. As shown in Figure 7A, the Ad-Flc group exhibited a decline in the serum estradiol level compared with the control group GFP (*p* < 0.05). The protein level of the ferritin light chain/heavy chain and transferrin increased in the 9th week of a bilateral intraovarian single-injection of Ad-Flc, Ad-Fhc, and Ad-Tf, respectively, (Figure 7B–E) compared with control group GFP, which confirmed that the adenovirus was effective in infecting the ovaries. The protein levels of chronic inflammation markers, including NF-κB p65 (Figure 7G,F), NF-κB p50 (Figure 7H,G), and iNOS (Figure 7I,H), increased significantly in all Ad-treated groups compared with the control group GFP. The protein level of GPX4 decreased in the Ad-Fhc group (*p* < 0.05) (Figure 7I,J) compared with the control group GFP. Therefore, the overexpression of the ferritin light chain/heavy chain and transferrin potentially induced the decline in the estradiol level via the up-modulation of NF-kB p65/50, which mediated the iNOS expression, and the down-modulation of the GPX4 expression in rat ovaries. Figure 8 shows the dose-dependent increase in the levels of the ferritin light chain after the treatment with increasing concentrations of ferritin for 12 h on primary granulosa cell cultures.

**Figure 7 ijms-23-12689-f007:**
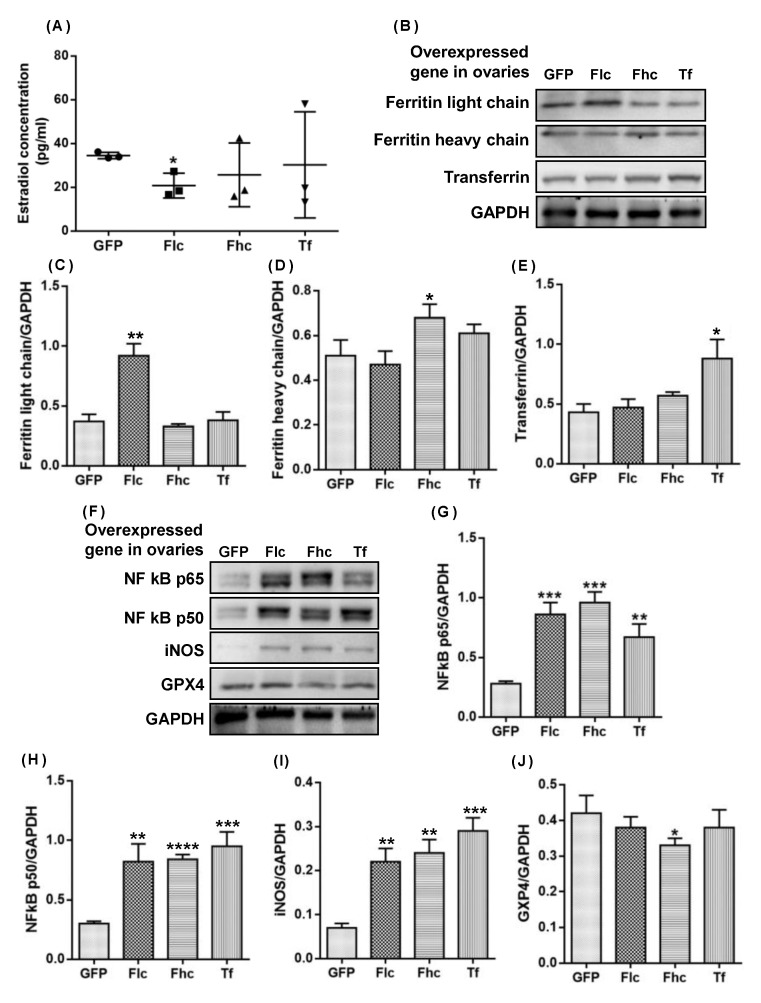
**Western blots and quantification of various proteins in ovaries of young rats with overexpressed ferritin light chain (Flc), ferritin heavy chain (Fhc), and transferrin (Tf) genes.** (**A**) Serum estradiol concentration, (**B**) Western blots and quantification of (**C**) ferritin light chain, (**D**) ferritin heavy chain and (**E**) transferrin (**F**) Western blots and quantification of (**G**) NF-κB p65, (**H**) NF-κB p50, (**I**) iNOS, and (**J**) GPX4 in rat ovaries with overexpressed ferritin light chain (Flc), ferritin heavy chain (Fhc) and transferrin (Tf) genes. Data are presented as mean ± SD (*n* = 3). The differences in protein expression were compared between rat ovaries and those overexpressing GFP gene as control using unpaired *t* test. *, *p* < 0.05; **, *p* < 0.01; ***, *p* < 0.001; ****, *p* < 0.0001.

**Figure 8 ijms-23-12689-f008:**
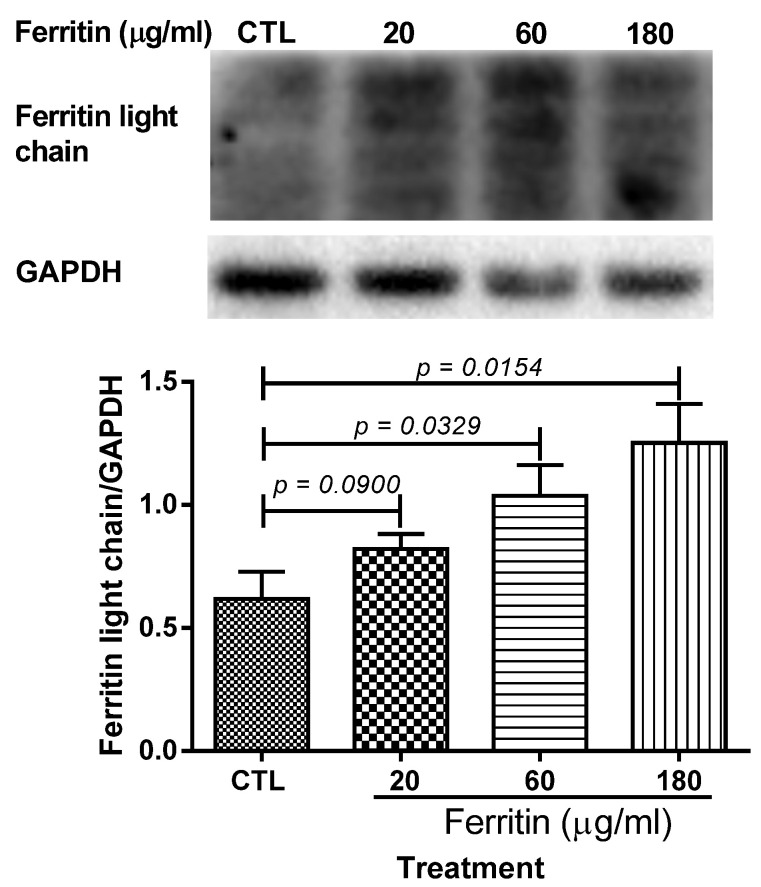
**The levels of ferritin light chain after treatment with different concentrations of ferritin for 12 h in primary granulosa cell cultures.** Data are presented as mean ± SEM (*n* = 3). The data between different groups were compared using one-tailed unpaired *t* test. CTL, blank control without ferritin supplement in the culture medium.

## 3. Discussion 

The current evidence indicates an association between Fe accumulation and menopausal symptoms [26,27]. We performed a proteomics analysis using Blast2Go and a naturally aging rat model to disclose a novel key pathway in ovarian aging involving the aberrant upregulation of Tf and ferritin Ft ensuing in the Fe accumulation in the ovaries, the aberrant upregulated FTH1, Ftl1, and Tf that induced a dysregulated Fe metabolism, and the upregulated I-κB kinase/NF-κB signaling that mediated the dysregulation of metabolic processes involving nitrogen compounds (Figure 1, Table 1, and Appendix A), including the upregulation of NO and 3-NT (Figure 3). NF-κB exists in the cytoplasm in an inactive complex bound to IκB and is activated via a pathway involving a phosphorylation-dependent, proteasome-mediated IκB breakdown. The pathway entails IκB kinase activation. 

Western blotting was used to examine the mechanism of age-associated Fe accumulation in naturally aging rat ovaries (Figure 1 and Figure 2) and to first demonstrate that the ovarian Tf and FTH/Ftl chain were aberrantly upregulated in these ovaries (Figure 2). The mechanism and impact of the iron metabolism in normal ovarian aging are distinctly different from those in polycystic ovary syndrome, although the latter shows iron-sparing chronic oligomenorrhea with an iron overload [28]. Tf and Ft are involved in the negative feedback regulation of Fe that maintains Fe homeostasis [12]. Their expression levels are normally inversely regulated in tissues [29]. An aberrant ovarian Tf and FTH/Ftl chain upregulation may contribute to an age-associated ovarian Fe accumulation (Figure 2). Tf can be expressed [30] and inhibits ovarian granulosa cell progesterone (P) production [31]. In the acute phase, Ft is an antioxidant converting Fe^2+^ ions into Fe^3+^ ions and stores inactivated iron as Fe^3+^ ions. In the chronic phase, Ft has a cytoprotective role. Additionally, a chronic H-ferritin overexpression may elicit an increase in the cytosolic labile Fe pool in mice, suggesting that Fe stored in Ft may eventually become a pro-oxidant. We found the upregulation of proteins associated with the Fe metabolism and aging ovaries involved the IRE/IRP regulatory circuit, and the upregulation of the ovarian TfR1 via the upregulated IRP2 in naturally aging ovaries in vivo. The TfR1 transports Fe into cells by binding to the Tf and Ft. The post-transcriptional IRE/IRP regulatory circuit plays an essential role in Fe homeostasis in various cell types. The IRP1 and IRP2 binding to IREs in the 5′-UTR of FTH1 and Ftl mRNA inhibit their translation, whereas IRP binding to the 3′-UTR of TfR1 mRNA prevents its degradation, which then promotes the TfR translation [32] and increases the TfR1 expression. The IRP1 expression and IRP2 expression exhibit different tissue specificities [12,33]. When there is more IRP2 in the cells, IRE-dependent repression mainly inhibits the translation of Ft rather than other mRNAs, which can explain the tissue-specific Fe regulation by Ft mRNAs [12,33]. This is the first report that upregulated IRP2 promotes TFR1 protein expression but does not inhibit the protein expression of the FTH1 and Ftl chains in the naturally aging ovary (Figure 2). The upregulated FTH chain may be regulated by the upregulated NF-κB at the transcriptional level (Figure 2 and Figure 5). Fe^2+^ binds to the 5′-UTR IREs’ stem-loop, changing its conformation, which reduces its affinity for IRPs but enhances its affinity for the ribosome recruitment factor eIF4F, promoting the FTH chain translation [34]. This implies that an age-associated ovarian Fe^2+^ accumulation (Figure 3D) may contribute to the aberrantly enhanced translation of Ft via 5′-UTR IREs, and thus the aberrantly upregulated Ft in the ovaries of aging rats. As aging proceeded, the ovarian iron level as well as the IRP2 protein expression level increased (Figure 2F and Figure 3I), in discord with the observation [35] that an Fe–S cluster within the FBXL5 (F box protein) enhances the IRP2 polyubiquitination and degradation in response to both Fe and OS. Nevertheless, the outcome of the relationship between the IRP2 and intracellular Fe level may differ between the different tissues [36]. *Irp2*−/− mice exhibited an Fe overload in the liver and duodenum but an Fe deficiency in the bone marrow and spleen. The conditional *Irp2* deletion in mouse liver and duodenal cells repeats the Fe overload phenotype of the liver and duodenum, indicating that the Fe overload is attributed to the cell-autonomous functions of Irp2 deficiency in these cells. Conditional *Irp2* deletion in splenic macrophages does not produce the splenic Fe deficiency seen in *Irp2*−/− mice, indicating that an Fe deficiency in *Irp2*−/− mouse spleen is secondary to Fe dysregulation in other cell types. 

The absolute ovarian weight and ovarian weight showed an age-associated decrease (Figure 3A,B). The ovarian weight in menopausal females decreased to half of that in young females [37]. The ovarian weight in 12-month-old rats was reduced to 53% of that in 9-month-old rats (Figure 3B) and further reduced in 16-month-old rats, which was not significantly different from that in 12-month-old rats (Figure 3B). This implies that rats may enter the equivalent of human menopause at or before the age of 12 months, well before the age of 15–18 months mentioned in [38]. Furthermore, the serum estradiol (E2) levels declined in 12-, 16-, and 22-month-old rats compared with 3- and 9-month-old rats. In humans, the serum E2 levels start to fall 2 years prior to menopause [39]. The iron, nitrate/nitrite, 3-NT, and HNE levels were upregulated in the ovaries of 12-month-old rats compared with 9-month-old rats (Figure 3D,J), suggesting that 12-month-old rats possibly enter the equivalent of human menopause, providing a model for studying ovarian aging. Thus, 9- to 12-month-old rats are preferred to 15- to 18-month-old rats for their use in screening drugs for prolonging the ovarian lifespan, delaying ovarian aging and menopause.

The Fe, nitrite/nitrate (a biomarker of nitric oxide), 3-NT, and HNE levels increased with age in the ovaries of 9- to 22-month-old rats (Figure 3D–J), consistent with [40]. Free/protein-bound tyrosines are acted on by RNI, to yield free/protein-bound 3-NT, a marker of oxidative stress. HNE, an alpha, beta-unsaturated hydroxyalkenal and a lipid peroxidation product in cells, triggers cell death and various diseases when tested at high levels. 

The non-heme Fe levels in mouse ovarian tissues were elevated at 7, 13, 46, and 63 weeks of age [9], indicating that Fe accumulation plays a role in aging. In patients with β-thalassemia major and idiopathic hemochromatosis, reproductive failure is attributed to a pituitary and gonadal Fe accumulation and dysfunction [41,42]. The total Fe and Ft concentrations in the follicular fluid of these patients were 6.7-fold and 53-fold higher than those in healthy human subjects [43]. The Fe content likely contributes to the ovarian weight increase in 9-month-old compared with 3-month-old rats (Figure 3B). Excess Fe induces disorders in physiological functions through ROS overproduction, leading to OS, lipid oxidation, and damage to tissues, cells, proteins and DNA [44]. The administration of an Fe^2+^-citrate complex to mice pretreated with lipopolysaccharide (LPS) increased LPS-induced nitric oxide (NO) formation in various organs by 10–20 fold [45]. NO is a mediator of iron-induced toxicity in primary proximal tubule cells [46]. NO-induced toxicity is independent of lipid peroxidation, which might explain the variable effects of different antioxidants on cell damage and lipid peroxidation in Fe-induced cytotoxicity. High NO levels can cause apoptosis and inhibit follicular steroidogenesis in vitro in human and mammals [47]. The mechanism involves the NO stimulation of O^2−^/H_2_O_2_/OH-induced lipid oxidation [48], and NO binding to aromatase and the inhibition of its activity in granulosa cells [47]. 

FeSO_4_ inhibited the IGF-I secretion and increased the caspase-3 expression in porcine ovarian granulosa cells. Hence, we tested the effect of Ft on granulosa cells [49] instead. Ft significantly curtailed E2 biosynthesis in ovarian granulosa cells in vitro via the upregulation of NF-κB p65/p50-induced oxidative and inflammatory factor iNOS (Figure 4). iNOS, which catalyzes the reactive nitrogen species (NO) formation, is upregulated by NF-κB that responds to cytokines, free radicals, stress, infection, and inflammation. NO reacts with superoxide to form peroxynitrite, an oxidant and nitrating agent which elicits cellular damage such as DNA damage and upregulates cell death pathways. Pavlová et al. demonstrated that p50/p50 stimulated proliferation and inhibited apoptosis in ovarian granulosa cells, whereas p65/p65 also stimulated proliferation but increased mitochondrial/Bax-related apoptosis [50]. NF-κB regulates ovarian cell function and pathological transformation involving its p65 (RelA) and p50 subunits [49,50]. Alcoholic liver disease is potentially induced by Fe via the activation of NF-κB p65/p50 in Kupffer cells [51]. Ft induces proinflammatory iNOS via p50/p65-NF-κB activation in the primary hepatic stellate cells in response to liver injury [52]. Fe/Ft can also induce NF-κB activation in mengovirus-infected mouse fibroblasts (L929 cells) and hepatic macrophages [53,54].

The NF-κB/iNOS signaling pathway was upregulated in aging rat ovaries (Figure 5) and in 3-month-old young rats following an intra-ovarian injection of an adenovirus expressing Ft and Tf (Figure 7), indicating that elevated ferritin and transferrin levels directly promote the ovarian NF-κB/iNOS signaling pathway with aging. The ovarian NF-κB p105 level increased with age [55]. NF-κB has dual roles in the regulation of cell survival and inflammatory pathogenic signaling pathways [56]. During activation, it binds to nuclear κB sites to regulate gene transcription. It is activated by proinflammatory cytokines, and by reactive oxygen species that induce the degradation and release of I-κB from p50/RelA (p65) and/or p50/c-Rel(Rel) to activate a cytokine gene expression and iNOS, whose proteins promote NF-κB activation in a positive feedback loop [56]. NF-κB can mediate pathogenic signaling pathways, leading to disease, cell degeneration, aging, and death [57]. The activated NF-κB pathway upregulates iNOS, promoting NO production, that induces cell death and decreases E2 biosynthesis. Excess NO also causes apoptosis and inhibits P production in rat granulosa cells [58]. The depressed ovarian weight, decreased E2 levels, and upregulated nitrite/nitrate and 3-nitrotyrosine (Figure 2) are possibly caused by the upregulated iNOS due to the activated NF-κB signaling (Figure 5). 

The Nrf2/GPX4 signaling pathway was downregulated with age in the ovaries of aging rats (Figure 6), indicating an abated antioxidant defense and heightened oxidative stress (Figure 3F–J) induced by Fe assimilation (Figure 3D,E) and the iNOS upregulation, mediated by the activated NF-κB (Figure 5A–J). The decline in the adaptive response of antioxidants to oxidative stimuli, such as the accumulation of oxidative damage with age, involves Keap1-Nrf2 signaling [59,60]. Keap1 facilitates the ubiquitination and subsequent proteolysis of Nrf2, a basic leucine zipper protein which regulates the expression of antioxidant proteins, upregulates phase II enzymes, and inhibits NLRP3 inflammasome. Age-dependent changes in basal Nrf2 and its activation in different tissues have been reported, but there is a lack of similar information relating to the ovaries and aging.

GPX4, an enzyme involved in defense and detoxification, protects phospholipids, cholesteryl esters, and cardiolipin; defends against apoptosis [61]; guards organisms from oxidative damage [62]. Gpx4-null mice (Gpx4−/−) die during early embryonic development and Gpx4+/− cells are vulnerable to oxidative stress [63,64]. Gpx4 downregulation induces iron-dependent ferroptotic cell death [65,66,67,68]. Ovarian GPX4 expression was downregulated in aging rats (Figure 6A,D) and in 3-month-old young rats in response to an intra-ovarian injection of an adenovirus expressing the ferritin-heavy chain (Figure 7G), which contributed to an attenuated antioxidant defense, leading to iron-induced ovarian aging. We found that the Keap1 upregulation induced the downregulation of Nrf2 and downstream Gpx4 in the ovaries of 9-, 12-, 16-, and 22-month-old rats (Figure 6A–D). We demonstrate a trend of downregulation in the protein expression level of the antioxidant enzyme Gpx1 in the ovaries of 3-, 9-, and 12-month-old SD rats, despite a slight upregulation in 16-, and 22-month-old SD rats (Figure 6A,E). The Gpx1 mRNA was upregulated in the ovaries with age in naturally aging mice [40]. Glutathione S- transferase Pi 1 (GSTP1), which plays a role in xenobiotic metabolism, was slightly upregulated with age in the ovaries of 3-, 9-, 12-, 16-, and 22-month-old SD rats (Figure 6A,F). Ovarian aging was associated with an undermined GSTP1 protein expression in follicular fluid samples from women aged 39–45 years compared with younger women aged 27–32 years [69], although no significant changes in the GSTP1 mRNA were detected in the ovaries of aging mice [40]. These findings and ours imply that changes in ovarian GSTP1 expression with age may be species-specific. We demonstrated in this study that aberrant transferrin and ferritin upregulation triggered an iron accumulation via the upregulation of IRP2-induced TfR1. Although the concomitant upregulation is at variance with the reports on other cells/organs, the findings are in keeping with those of Byrd and Horwitz [70] who showed that the transferrin receptor expression and intracellular ferritin content in human monocytes were enhanced by iron transferrin and reduced by interferon-gamma. Under the influence of Alpha1-antitrypsin, the synthesis of the ferritin as well as transferrin receptor, and the cellular contents of the ferritin H-chain mRNA and transferrin receptor mRNA were upregulated [71]. Kobak et al. [72] demonstrated an augmented expression of the ferritin and transferrin receptor 1 in primary human cardiofibroblasts and cardiomyocytes treated with sera from patients with acute myocarditis compared to the cells treated with sera from healthy control subjects.

Fe plays many crucial roles in the body. The maintenance of a normal Fe level is important to people of different ages, including the elderly. Therapeutic measures to tackle Fe deficiency and Fe overload are of utmost importance. An iron overload is associated with a myriad of ailments of different organs and systems. An acute Fe overload leads to hypothalamic-pituitary-ovarian axis abnormalities in rats [73]. A Tf deficiency and Fe overload in follicular fluid account for oocyte dysmaturity in infertile patients with advanced endometriosis [74]. A reduced reproductive ability is seen in thalassemia major patients with an Fe overload [41]. HNE [75] and ROS [76] contributes to the OS-mediated deterioration of an aging oocyte. The NT and HNE levels and the number of mitochondria showing membrane damage in mouse ovarian cells were enhanced, whereas P and testosterone levels, ovarian reserves, fertility, and fecundity declined. An excessive storage of non-heme Fe in ovarian stromal tissue may be associated with ovarian aging due to an elevated OS [77]. Nulliparity brings about ovarian Fe accumulation and the senescence pigment lipofuscin as well as oxidative damage to DNA in mice. Lipofuscin-associated Fe gives rise to ROS in aged cells [78]. Oxidative stress, usually associated with mitochondrial dysfunction, results in the apoptosis of ovarian cells. Mitochondrial dysfunction and a decrease in the mitochondrial DNA copy number are seen in the aging of different tissues, including the ovary. Therapeutic strategies directed to the mitochondria would be useful to improve fertility. Menopause is a terminal stage in ovarian aging. The age-associated decline in the number of ovarian follicles determines the start of menstrual cycle irregularity and the final cessation of menstruation. Oocyte quality deterioration accounts for a progressive loss in fertility. The abated negative feedback by a reduced titer of estradiol at the hypothalamic and adenohypophyseal levels ensues in the heightened levels of the follicle stimulating hormone. The level of anti-Mullerian hormone plummets. Previous and the present research disclosed that ovarian aging mechanisms are multifaceted and involve oxidative stress, an Fe overload, and endocrine factors. The elucidation of the mechanisms will help predict menopause and prolong the time span of fertility [79]. Compared to the control group without an Ft supplement, the levels of the Ftl chain in the granulosa cells were higher after treatment with 20, 60, and 180 μg/mL of Ft (*p* value = 0.090, 0.0329, and 0.0154, respectively). A correlation analysis of the levels of Ftl and Ft treatments showed a significant positive correlation (r = 0.9325, *p* = 0.0338, one-tailed Pearson correlation coefficient). A correlation analysis of the E2 levels and Ftl levels exhibited a significant negative correlation (r = −0.986, *p* = 0.007, one-tailed Pearson correlation coefficient). Thus, the Ftl chain had inhibitory effects on E2 levels in the primary granulosa cell culture medium, indicating a suppressive action of the Ftl chain on E2 biosynthesis in the ovarian granulosa cells. This is consistent with our in vivo study, demonstrating that an ovary-specific over-expression of the Ftl chain in the young ovaries, the major organ of E2 biosynthesis, could down-regulate the serum E2 levels without a modulation of the levels of Tf and the FTH chain (Figure 2 and Figure 3). In addition, the FTH chain and the iron in the complex possibly had inhibitory effects on E2 biosynthesis. An ovary-specific over-expression of the FTH chain in the young ovaries also down-regulated the serum estradiol level without a significant modulation of the levels of transferrin and the ferritin light chain. It was reported that 50 μmol/L (2.79 μg/mL) of Fe2+ iron led to a small insignificant inhibition of the activities of aromatase, the key enzyme for estradiol biosynthesis, in human placental microsomes [80], which implied that iron possibly possessed small insignificant inhibitory effects on E2 biosynthesis. Moreover, although 500 μmol/L (27.9 μg/mL) of Fe^2+^ iron could induce cell arrest in the mouse granulosa cell cultures, both 500 μmol/L (27.9 μg/mL) of Fe^3+^ iron and 100 μmol/L of Fe^2+^ (5.58 μg/mL) did not lead to cell cycle arrest [81]. Treating the porcine ovarian granulosa cells with 0.17 and 1.0 mg/mL of Fe^2+^ iron supplement had no effect on the secretion of progesterone, an E2 precursor [49]. Thus, the incubation of free Fe^3+^ on primary ovarian granulosa cells possibly caused slight insignificant toxic effects, which were largely lower than the toxic effects induced by the incubation of the free Fe^2+^ iron. 

Senescent cells are characterized by an Fe overload caused by impaired lysosomal ferritinophagy, in which Fe is sequestered in Ft and has a resistance to ferroptosis. Rapamycin enhances Ft breakdown and inhibits the elevation of TfR1, ferritin, and intracellular iron [82]. An Fe overload, especially if it takes place in the mitochondria, enhances redox-active iron availability, which may lead to aging-associated oxidative damage and, in turn, diseases [83]. A chronic iron overload upregulated the inducible nitric oxide synthase (iNOS) expression, enhanced the NF-kappaB DNA binding to the iNOS gene promoter, and upregulated the expression of the iNOS mRNA and iNOS activity at 8 and 12 weeks [84]. Iron exacerbates hepatic oxidative stress in humans and animals, which may trigger the expression of redox-sensitive genes. An iron overload induced by ferric hydroxide polymaltose produced an elevation in the circulatory Fe, Ft, Tf, Tf saturation%, total Fe binding capacity, and aminotransferase activities, and also enhanced thr hepatic Fe, lipid peroxidation marker malondialdehyde (MDA), and NO levels, but reduced the activities of antioxidative enzymes [85]. The heightening of blood Ft concentration indicating body Fe stores is seen during inflammation in chronic hepatic damage. Ft exhibits proinflammatory activity through signaling, mediated by NF-kappaB in hepatic stellate cells [86]. Fe liberated from Ft- and NO-derived radicals may aggravate OS [87].

Nonheme Fe (hemosiderin) and the senescence marker lipofuscin were observed in aging murine ovaries [78,88,89]. Multinucleated giant cells derived from macrophages in these ovaries indicates inflamm-aging [90], which is linked to premature ovarian insufficiency [91]. The EGR1 upregulation exerts antiproliferative- and apoptosis-inducing activities toward granulosa cells during follicle atresia in the aging murine ovary [92]. A chronic low-grade systemic inflammation brought about by the NLRP3 inflammasome results in the reduction in ovarian follicles in the aging ovary [93]. In the dehydroepiandrosterone-induced mouse model of polycystic ovary syndrome, ferric salt stimulated TFRC which augmented the Fe content, brought about ROS liberation, stimulated mitophagy, and triggered lipid peroxidation, further promoting the ferroptosis of KGN human granulosa-like tumor cells. The suppressive action of TfR/NADPH oxidase 1/PTEN, which induced kinase 1 acyl-CoA synthetase long chain family member 4 signaling on folliculogenesis, is a likely target for polycystic ovary syndrome [94]. Further to the E2 decline, Fe accumulation due to menopause constitutes a risk factor by increasing the sensitivity of skin fibroblasts and keratinocytes to UVA [95]. 

IRP1 and IRP2 regulate the production of Ft and TfR through binding to IREs in the 3’ untranslated region (UTR) and the 5’ UTR of the respective mRNAs. The Fe levels in the cell influence the binding of IRPs to IREs, and hence Ft and TfR expression. NO*, a NO redox species which shows interactions with Fe, upregulates the IRP1 RNA-binding activity, leading to increased TfR mRNA levels. The exposure of murine macrophage RAW 264.7 cells to NO+ ion, which S-nitrosylates SH groups, ensued in a decreased IRP2 RNA binding, followed by IRP2 breakdown, and reduced TfR mRNA levels. IFN-γ and LPS stimulated the RAW 264.7 cells and enhanced the IRP1 binding activity, but the IRP2 RNA binding diminished, followed by an IRP2 breakdown, and a decline in the TfR mRNA levels. iNOS inhibitors inhibited these changes, indicating that an NO+-mediated breakdown of IRP2 is important in the Fe metabolism during inflammation [96]. Posttranscriptional control of Fe homeostasis in the cell is mediated by IRP1 and IRP2. In the absence of Fe, IRPs bind to IREs in the 5’ untranslated region of ferritin mRNA and the 3’ untranslated region of transferrin receptor mRNA. IRP binding to the IREs inhibits Ft translation and enhances the TfR mRNA stability, while the contrary takes place in iron- replete cells. IRP binding to IREs is influenced by NO. A brief treatment of RAW 264.7 macrophage cells with sodium nitroprusside (SNP), an NO+ donor, markedly elevated Ft biosynthesis which was inhibited by MG132, which also prevented an SNP-induced the IRP2 breakdown. Exposure of RAW 264.7 cells to LPS and IFN-γ brought about the IRP2 breakdown and upregulated Ft biosynthesis, which were inhibitable by iNOS inhibitors. An SNP-elicited rise in Ft biosynthesis was accompanied by an increased Fe incorporation into Ft. Thus, NO+-mediated modulation of IRP2 plays a critical role in regulating Ft biosynthesis and the Fe metabolism in murine macrophages [97]. OS provokes apoptosis, inflammation, mitochondrial dysfunction and telomere shortening, which are all associated with ovarian aging. Natural antioxidants protect the ovaries [98]. NO generated by iNOS exerts inhibits E2 production by ovarian granulosa cells and P production by luteal cells, and also aromatase activity [99]. We show that the upregulation of Fe, Ft and Tf, iNOS, RNI, NFkB, and TNF-α, and the downregulation of ovarian steroids and antioxidative enzymes in the aging ovary, resemble that in other organs. Our findings are important in view of the great impact of ovarian aging on female physiology and will lay a good foundation on subsequent research to combat ovarian aging.

Meyron-Holtz et al. [100] compared the consequences of genetic ablation of IRP1 to that of IRP2 in mice. The dysregulation of the iron metabolism is discernible in all tissues in IRP2−/− mice, but takes place only in brown fat and the kidneys in IRP1−/− mice. The IRP2 exhibits a sensitivity to the iron status and can make up for the loss of IRP1 by elevating its binding activity. On the other hand, the small RNA-binding fraction of the IRP1, which lacks sensitivity to the cellular iron status, plays a role in basal mammalian iron homeostasis. Hence, IRP2 has a central importance in the post-transcriptional regulation of the iron metabolism in mammals. Schalinske et al. [101] noted that the Ba/F3 family of murine pro-B lymphocytes IRP1 is not crucial for the regulation of ferritin or TfR expression by iron and that IRP2 can act as the sole IRE-dependent mediator of cellular iron homeostasis. Thus, IRP1 and IRP2 play different roles in the regulation of the iron metabolism. In our study, IRP2 played a more pronounced role than IRP1 (Figure 2F,H,I).

Bisphenol A modified the expression of the ferroptosis-related genes, ferritin heavy chain 1 (FTH1), and glutathione peroxidase 4 (GPX4) in mouse testicular tissues [102]. Tetramethyl bisphenol A attenuated the expression of the ferroptosis-related genes, (Cat, Keap1, SOD1, and SOD2) and reduced the levels of ferroptosis pathway proteins (GPX4, Keap1, and NRF2) in Leydig cells [103]. Busulfan triggered ferroptosis in spermatogenic cells by down-regulating the expression of GPX4 and Nrf2 [104]. The transferrin receptor has been shown to be a specific marker of ferroptosis [105], and transferrin a regulator of ferroptosis [106]. Both the IRP1 and IRP2 enhance ferroptosis [107,108]. Our study demonstrated that all the aforementioned proteins, and also 4-HNE, which is a product of lipid peroxidation, play a pronounced role in ovarian aging, as has previously been reported for the testis, signifying the involvement of ferroptosis in ovarian aging in the rat.

In summary, we observed that an aberrant Tf and Ft upregulation in the ovaries of aging rats resulted in Fe accumulation and inflamm-aging via NF-κB-induced iNOS. An acute upregulation or low Tf and Ft levels exert beneficial effects, but a chronic upregulation produce toxicity, caused by Fe accumulation. Although the toxicity of Fe in extragonadal tissues can be relieved by exogenous E2 [23], Fe accumulation leads to reduced endogenous E2 biosynthesis in naturally aging ovaries. Since 9- to 12-month-old rats displayed menopausal characteristics, we recommend using them instead of the commonly utilized 16- to 18-month-old rats to screen drugs for prolonging the ovarian lifespan and delaying ovarian aging. Our findings add to the development of the theory of ovarian aging and the Fe metabolism and should stimulate translational and clinical investigations on normal ovarian aging and the associated diseases.

## 4. Materials and Methods

**Chemicals, antibodies, and vectors.** CHAPS (17-134-01), IPG buffer (*v*/*v*, 17-6000-88), dithiothreitol (DTT) (17-1318-02), iodoacetamide (RPN6302V), and Amersham ECL Advance Western Blotting Detection Kit (#RPN2236) were purchased from GE Healthcare, Chalfont Saint Giles, UK. Gelatin from porcine skin (G1890), silver nitrate (S8157), trypan blue (T8154), radioimmunoprecipitation assay (RIPA; R0278), penicillin-streptomycin (PS), and insulin were obtained from Sigma-Aldrich, Saint-Louis, MI, USA. The DMEM/F12 1:1 medium (SH3002.01) was from Hyclone, Thermo Scientific, Waltham, MA, USA. The phosphate-buffered saline (PBS) with a pH of 7.4 (10010023) was from Gibco, Life Technologies, Grand Island, NY, USA. The pregnant mare serum gonadotropin (PMSG) (367222) was from Calbiochem, San Diego, CA, USA. The protein assay buffer (#500-0006) was from Bio-Rad, Hercules, CA, USA. The protease inhibitor (sc-29131) was from Santa Cruz, CA, USA. The protease inhibitor cocktail tablet (04093159001) and estradiol II kits (#03000079) were from Roche Diagnostics, Mannheim, Germany. Ferritin (Sigma-Aldrich catalogue number F4503-25MG, CAS number 9007-73-2), with an iron content of 144 to 188 μg/mg, was isolated and purified from equine spleen and a natural complex comprised of H and L chains and a core with up to 4500 Fe^3+^ iron ions.

The primary antibodies against transferrin: (sc-22597), ferritin light chain (sc-14420), ferritin heavy chain (sc-14416), TfR1 (sc-9099), IRP1 (sc-14216), IRP2 (sc-14221), NF-κB p65 (sc-8008), NF-κB p50, iNOS (sc-7271), A20 (sc-166692), I-κB alpha (sc-1643), TNF alpha (sc-1351), TNFR1 (sc-8436), Nrf2, Keap1, GPX1 (sc-22146), and GPX4 (sc-166570) were obtained from Santa Cruz, USA. The primary antibody against ferroportin (AIT-001) was a free sample from Alomone labs, Jerusalem, Israel. The antibody of 3-Nitrotyrosine (3-NT) (5412-100) was from BioVision, Milpitas, CA, USA. The antibody of 4-Hydroxynonenal (HNE) (MAB3249) was from R&D Systems, Minneapolis, MN, USA. The anti-GSTP1 (GTX112695) antibody was from GeneTex, Irvine, CA, USA. The primary antibody against GAPDH (#2118) and the secondary antibodies anti-rabbit IgG (#7074) and anti-mouse IgG (#7076) were from Cell Signaling Technology, Danvers, MA, USA. The secondary antibody anti-goat IgG (AP106p) was from Millipore, Burlington, MA, USA.

The adenovirus used for the ovarian injection was generated by the Vigene Bioscience (Jinan, Shandong, China). A typical of Ad-Ftl1 (1.9 × 108 PFU/mL), ad-Fth1 (1.1 × 108 PFU/mL), and ad-Tf (1.2 × 108 PFU/mL) was obtained. The empty adenoviral vector (1.0 × 108 PFU/mL) was prepared in a similar fashion to serve as a vehicle control.

**Animal husbandry and ovary collection.** The experiments had been approved by the Committee on the Use of Live Animals in Teaching and Research (CULATR Ref. 3203-14) of Li Ka Shing Faculty of Medicine, the University of Hong Kong, and Faculty Research Grant Committee (REC/18-19/0256), Hong Kong Baptist University prior to the commencement of the experiments. Naturally aging female SD rats were used as the animal model for the mechanistic study of ovarian aging. Animals aged 3 and ~8 months were purchased from the Laboratory Animal Unit of the University of Hong Kong. Animals were housed in 12 h light/12 h dark cycles at an ambient temperature of 24 °C and a relative humidity of 50–65% until the ages of 3, 9, 12, 16, and 22 months, which represent the periods of early reproductive maturity, active reproductive age, late reproductive age, post-reproductive age, and senescence, respectively. Rats at the required ages (3, 9, 12, 16, and 22 months) with a body condition score of 3 were weighed and euthanized with an intraperitoneal injection of 100 mg/kg ketamine-xylene (2:1, *v*/*v*). The harvested ovaries were weighed and washed with PBS, and quickly frozen in liquid nitrogen and stored at −80 °C for further experiments.

Three-month-old female young SD rats were assigned randomly into 4 groups, including an ad-Ftl group, an ad-Fth group, an ad-Tf group, and a control group with 3 rats in each group. The rats were anesthetized with ketamine (100 mg/kg) and xylazine (10 mg/kg) *i.p.* After laparotomy, all ovaries of each group were injected with Ad-Ftl, Ad-Fth, Ad-Tf, and an empty adenoviral vector (with GFP labeling, 5 μL for each side), respectively, by using a Hamilton syringe (Harvard Apparatus, Holliston, MA, USA), and then returned in the pelvis. The fascia was closed using interrupted sutures and/or the skin stapled. Sterile techniques were used throughout the course of the experiment. After surgery, the animals were allowed to recover for 9 weeks. Subsequently, the blood was collected from the caudal vein during the estrus phase for E2 measurement, and the ovaries were collected for Western blotting analysis of NFkB/iNOS and GPX4.

**Proteomics analysis of aging ovaries.** The ovarian tissues (~50 mg) from each group were ground into powder in liquid nitrogen. The proteins were extracted in 500 μL lysis buffer [8 mol/L urea, 4% CHAPS (*w*/*v*), 2% IPG buffer (*v*/*v*), 40 mmol/L DTT in double-distilled water] with 10 μL of 20 mmol/L PMSF, 3.3 μg/μL of leupeptin, and 3.3 μg/μL of aprotinin. The protein extracts were sonicated on ice for 15 min and centrifuged (15,700× *g*) for 30 min at 4 °C. The protein supernatant was purified with a 2D clean up kit and quantified with a 2D Quant kit. Isoelectric focusing (IEF) was performed using 17 cm non-linear ReadyStripTM IPG strips (pH 3–10) (Bio-Rad) at 20 °C using a Bio-Rad Protean IEF Cell (Bio-Rad). The proteins (120 μg) were mixed with a rehydration buffer (8 M urea, 2% CHAPS, 2.8 mg/mL dithiothreitol, 0.5% IPG buffer, and 0.002% bromophenol blue) and loaded onto 1D IPG gel strips. The IPG strips were rehydrated with samples at 50 V for 12 h. The following IEF program was used: 150 V for 2 h; 1000 V for 0.5 h; 4000 V for 0.5 h; 8000 V for 30,000 Vh; and 10,000 V for 30,000 Vh, for a total of 62,825 Vh. After the completion of IEF, the strips were equilibrated for 15 min in an equilibration buffer [6 M urea, 75 mM Tris-HCl (pH 8.8), 29.3% glycerol, 2% SDS, 0.002% bromophenol blue] with 10 mg/mL dithiothreitol and then in equilibration buffer with 25 mg/mL iodoacetamide.

Next, 2D SDS-PAGE was performed in a PROTEAN II XL system (Bio-Rad) in 12% SDS-PAGE gel at 4°C at 100 V for 1 h followed by 250 V until the bromophenol blue dye reached the bottom of the gel. The gels were fixed in a 40% methanol and 10% acetic acid solution for 30 min, and then sensitized in 30% ethanol, 4.05% sodium acetate, and 0.2% sodium thiosulphate. After washing three times for 10 min each time in Milli-Q water, the gels were incubated in 0.1% silver nitrate solution for 40 min. The gels were washed in Milli-Q water and developed in a solution containing 2.5% sodium carbonate and 0.02% formalin for 15 min. The reaction was stopped with 1.46% EDTA solution, and the gel was washed three times for 5 min in Milli-Q water.

The gel images were analysed using PDQuest 8.0 2D Analysis Software (Bio-Rad) in accordance with the PD Quest Manual. Spots that showed more than a 3-fold change in % intensity x area spot quantity in 16- and 22-month-old SD rats compared with 3-month-old SD rats were selected and cut into small pieces. The spots were destained in destaining solution (15 mM potassium ferricyanide and 50 mM sodium thiosulfate), washed twice for 10 min each time in Milli-Q water, and then twice for 15 min each time in 50% acetonitrile. The digested peptide spots were identified with an ABI4800 MALDI TOF/TOF™ Analyzer at the Centre for Genomic Sciences, the University of Hong Kong. A functional analysis was performed on the identified proteins based on protein ontology (GO) categorization with Blast2GO software (https://www.blast2go.com, accessed in January 2018), including annotation, mapping, and InterPro analyses. GO analysis was carried out at level 2 to 7 for the 3 main categories (cellular component, molecular function, and biological process) with cutoffs of 5, 5, and 3 for protein distribution, respectively.

**Measurements of labile ferrous iron, total iron, and nitrate/nitrite levels in ovaries of naturally aging rats.** Ovarian tissue (~15 mg) was homogenized in 150 μL of ice-cold PBS (pH 7.4) and then centrifuged at 10,000 g for 20 min at 4 °C. The supernatant was retained and used for the determination of the protein content, iron levels, and nitrate/nitrite levels. The protein concentration was determined by Bradford assay (Bio-Rad) using a microplate reader (Bio-Rad) at 595 nm. The ovarian iron level was measured by the QuantiChromTM Iron Assay Kit (DIFE-250, BioAssay Systems, Hayward, CA, USA) according to the manufacturer’s instructions. Briefly, 25 μL of the sample in a 96-well plate was mixed with 100 μL of working reagent without reductant to determine the ferrous iron, which specifically binds to the chromogen to form a blue complex. For the determination of total iron (heme and non-heme iron), the working reagent contained a reductant that catalyzes the reduction of ferric iron to ferrous iron. After incubation for 40 min, the ferrous iron and total iron concentrations were measured at 595 nm using a microplate reader and expressed as ng iron/mg protein.

For nitrate/nitrite detection, the supernatant was ultrafiltered in a Nanosep 10K Omega Centrifugal Device (P/N OD010C33, Pall Corporation, Port Washington, NY, USA) at 14,000× *g* for 20 min at 4 °C. The nitrate/nitrite level was quantified using a nitrate/nitrite colorimetric assay kit (78,001, Cayman Chemical, Ann Arbor, MI, USA) following the manufacturer’s instructions. Briefly, 40 μL of filtered supernatant was diluted with 40 μL of assay buffer in a 96-well solid plate, and 10 μL of nitrate reductase cofactor mixture and nitrate reductase mixture were added and incubated for 3 h to catalyze the conversion of nitrate to nitrite. Next, 50 μL of Griess reagent R1 and R2 were added to each well sequentially and the mixture was incubated at room temperature for 10 min for color development. The total nitrate/nitrite level was determined at 540 nm using a microplate reader and expressed as nmol/mg protein.

**Measurement of serum estradiol levels.** To determine the circulating levels of ovarian hormones during the aging process, the serum estradiol levels were measured using a commercially available electro-chemiluminescence immunoassay kit, Estradiol II (#03000079, Roche Diagnostics) using an Elecsys 2010 autoanalyzer (Roche Diagnostics).

**Western blot analysis of proteins.** The ovarian tissue (~15 mg) was mechanically homogenized in 150 μL of RIPA buffer with a protease inhibitor (25 mL/tablet; Santa Cruz) and 1 mM phenylmethylsulsonyl fluoride (a serine protease inhibitor) to extract the proteins. The protein extracts were centrifuged at 15,700× *g* for 20 min at 4 °C, and the protein concentration was determined by using Bradford assay. A total of 20 μg of protein extracted from each ovary was denatured at 95 °C for 15 min. The proteins were separated by 12% SDS-PAGE with GAPDH as the housekeeping protein for the normalization.

After a transfer to a PVDF membrane, a blocking solution with 0.5% (*m*/*v*) gelatin in 0.1% TBS-T (Tris-buffered saline/Tween-20, *v*/*v*) was added for 1 h at room temperature. The membrane was incubated with primary antibodies overnight at 4 °C, washed three times for 10 min each time in 0.1% TBS-T, and then incubated with horseradish peroxidase-conjugated secondary antibodies for 1 h at room temperature. The chemiluminescence signal was detected using an Amersham ECL Advance Western Blotting Detection Kit (GE-Healthcare) and analyzed in a ChemiDoc EQ system (Bio-Rad).

**Primary culture of ovarian granulosa cells and ferritin treatment.** Approval for the experiments was obtained from the Department of Health, Hong Kong SAR and the Committee on the Use of Live Animals in Teaching and Research (CULATR ref. 3203-14), the University of Hong Kong. To study the mechanism of the depressed estradiol level due to the chronic iron accumulation, the ovarian granulosa cells were treated with ferritin to mimic the co-overexpression of a ferritin heavy chain and light chain. This ferritin treatment best mimics the physiological conditions, as ferritin can enter cells via the transferrin receptor 1. The isolation of ovarian granulosa cells and the cell culture followed our previous study, but with slight modifications. Briefly, the ovaries were collected from ~21-day-old female SD rats primed with PMSG for 48 h. The granulosa cells were isolated by puncturing the individual ovarian follicles with a 25-gauge needle and centrifuged at 250× *g* for 3 min. The cells were washed three times in PBS and a culture medium, and an aliquot of the cells was mixed with trypan blue stain to determine the cell number and viability. The cells were cultured in a 48-well plate at a cell density of 2 × 10^4^ cells/well in serum-free DME/F12 1:1 medium supplemented with 1% PS, 0.1% BSA, and 1 μg/mL of insulin together with the ferritin treatment (0, 20, 60, and 180 μg) for 12 h. The cultures were maintained at 37 °C in 5% CO_2_ in a cell culture incubator (Thermo Scientific Forma CO_2_). The cellular protein was extracted for the Western blot analysis and the cell culture medium was collected to determine the estradiol concentration.

**Detection of ovarian NF-κB binding activity in naturally aging rats.** Nuclear proteins were extracted from the ovarian tissues of SD rats (3, 9, 12, 16, and 22 months old) using NE-PER™ Nuclear and Cytoplasmic Extraction Reagents (#78833, Thermo Scientific) in accordance with the manufacturer’s instructions. The protein concentration was determined by Bradford assay. The electrophoretic mobility shift assay was performed using the Light Shift Chemiluminescent EMSA Kit (#20148, Thermo Scientific) according to the manufacturer’s instructions. NF-κB consensus double-stranded oligonucleotides (5′-AGTTGAGGGGACTTTCCCAGGC-3′) were produced by annealing complementary pairs of oligonucleotides labeled by biotin at the 5′ end (BGI, Shenzhen, China). The binding reaction contained binding buffer, 50 ng/μL poly (dI-dC), 5 mM MgCl_2_, 2.5% glycerol, 0.05% NP-40, 10 fmol of biotin-labeled double-stranded oligonucleotide, and nuclear or cytosolic extract (~2 μg of protein). Specificity was determined by a competition with a 200-fold excess of a non-biotin-labeled probe. Each reaction mixture was incubated for 20 min at room temperature. The samples were then subjected to electrophoretic separation at room temperature on a 7% native polyacrylamide gel at 100 V for 40 min.

After electrophoresis, the gels were transferred onto a nylon membrane at 100 V for 45 min. The membrane was crosslinked with UV light (254 nm) for 10 min using a hand-held UV lamp. Finally, the signals were detected by an Amersham ECL Advance Western Blotting Detection Kit (GE-Healthcare) using a ChemiDoc EQ system (Bio-Rad).

**Statistical analysis.** Data were collected from two independent experiments and presented as mean± SD. The differences between the groups were tested by a one-way ANOVA followed by Tukey’s multiple comparison test. The differences between the two age groups were compared using unpaired *t*-test.

## Figures and Tables

**Table 1 ijms-23-12689-t001:** Aging-associated changes in the expression of proteins in rat ovarian tissue revealed by proteomics analysis (data in bold face are related to iron storage or iron transport).

Protein Name	Accession No. (UniProt)	Protein Score	Protein Score C.I.%	Total Ion Score	Total Ion C.I.%	Protein MW	Protein pI	Pep. Count	Protein Function	Fold Changes (Compared to 3-Month-Old Group)		
										16-Month-Old	22-Month-Old	Mean
**1. Ftl1 (Ferritin light chain 1)**	**P02793**	**610**	**100**	**463**	**100**	**20,793**	**5.99**	**13**	**iron storage**	**5**	**5.67**	**5.33**
2. Cbr1 (Carbonyl reductase [NADPH] 1)	P47727	305	100	204	100	30,844	8.22	12	Antioxidative	−3.29	−3.83	−3.56
3. Fabp3(Fatty acid-binding protein)	P07483	422	100	330	100	14,766	5.9	9	lipid transport	3.44	3.39	3.42
4. Ldhb (L-lactate dehydrogenase B chain)	P42123	190	100	131	100	36,874	5.7	9	Oxidoreductase in the process of glycolysis	3.71	2.86	3.29
**5. Tf (Transferrin)**	**P12346**	**319**	**100**	**133**	**100**	**78,513**	**7.14**	**22**	**iron transport**	**3.43**	**3**	**3.21**
6. Phb (Prohibitin)	P67779	93	100	48	99.975	29,859	5.57	7	inhibition of cell proliferation	3	3.4	3.2
7. Gstt3 (Glutathione S-transferase theta-3)	D3Z8I7	307	100	217	100	23,822	9.3	10	combination of glutathione and oxidants	−3.09	−3.09	−3.09
8. Selenbp2 (Selenium- binding protein 2)	Q8VIF7	238	100	103	100	53,069	6.1	17	sensing of reactive xenobiotics in the cytoplasm, intra-Golgi protein transport	3.17	3	3.08
9. Hspa5 (78 kDa glucose-regulated protein)	P06761	466	100	332	100	72,474	5.07	20	molecular chaperone for protein folding	3.25	2.88	3.06
**10. Fth1 (Ferritin heavy chain)**	**P19132**	**328**	**100**	**238**	**100**	**21,284**	**5.85**	**10**	**iron storage**	**3**	**3**	**3**
11. Hba1 (Hemoglobin subunit alpha-1/2)	P01946	400	100	334	100	15,490	7.82	7	oxygen transport	3	3	3

## Data Availability

The data that supports the findings of this study are available from the corresponding author upon reasonable request.

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
