# Peer review of "Aberrant Transferrin and Ferritin Upregulation Elicits Iron Accumulation and Oxidative Inflammaging Causing Ferroptosis and Undermines Estradiol Biosynthesis in Aging Rat Ovaries by Upregulating NF-Κb-Activated Inducible Nitric Oxide Synthase: First Demonstration of an Intricate Mechanism"

_ijms, 2022, doi:10.3390/ijms232012689_

Round 1

Reviewer 1 Report (Previous Reviewer 2)

The work studies the ovaries of young and aged rats. It finds that ferritins, transferrin and iron are higher in the aged rats. Estradiol decreased with ageing, while indices of oxidative stress increased with upregulation of NF-kB and deregulation of iNOS. Ferritins were experimentally induced in primary granulosa cells and by adenovirus injections. The text is far too long and difficult to read. Some experiments are not convincing.

- The unexpected finding that both ferritins and TfR1 are upregulated during iron accumulation is in contrast with the notion they are controlled by IRPs in opposite directions. This should be analyzed more in detail, and the upregulation of TfR1 should be verified.

- The discussion is far too long, 400 lines!!!! It should be cut to half or less, and it should focus on the comments of the results, rather than being a review of iron metabolism. Also, a bibliography of about 200 references is far too long for a research article.

- There are a number of typos, and the bibliography is partly quoted by number and partly by author names. 

Author Response

Response to reviewer 1:

English language and style

(x) Extensive editing of English language and style required  

Authors’ response:

Language polishing of the manuscript has been performed by a professor in the Dept of English

Comments and Suggestions for Authors:

Q1. The work studies the ovaries of young and aged rats. It finds that ferritins, transferrin and iron are higher in the aged rats. Estradiol decreased with ageing, while indices of oxidative stress increased with upregulation of NF-kB and deregulation of iNOS. Ferritins were experimentally induced in primary granulosa cells and by adenovirus injections. The text is far too long and difficult to read. Some experiments are not convincing.

Authors’ response: The text has been cut short and improved by a professor in  the Dept of English

Q2. The unexpected finding that both ferritins and TfR1 are upregulated during iron accumulation is in contrast with the notion they are controlled by IRPs in opposite directions. This should be analyzed more in detail, and the upregulation of TfR1 should be verified. 

Authors’ response: An explanation has been added to lines 352-358 of DISCUSSION as follows.

Nevertheless, the outcome of relationship between IRP2 and intracellular Fe level may differ between different tissues [57]. Irp2-/- mice exhibited Fe overload in the liver and duodenum but Fe deficiency in the bone marrow and spleen. Conditional Irp2 deletion in mouse liver and duodenal cells repeats the Fe overload phenotype of the liver and duodenum, indicating that Fe overload is attributed to cell-autonomous functions of Irp2 deficiency in these cells. Conditional Irp2 deletion in splenic macrophages did not produce the splenic Fe deficiency seen in Irp2-/- mice, indicating Fe deficiency in Irp2-/- mouse spleen is secondary to Fe dysregulation in other cell types. 

Q3. The discussion is far too long, 400 lines!!!! It should be cut to half or less, and it should focus on the comments of the results, rather than being a review of iron metabolism. Also, a bibliography of about 200 references is far too long for a research article. 

Authors’ response:  The length of DICUSSION has ben reduced from 400 lines to 260 lines, and the number of references has been reduced from about 200 to 162.

Q4. There are a number of typos, and the bibliography is partly quoted by number and partly by author names. 

Authors’ response: The typographical errors have been corrected. The references are now all quoted by number

Reviewer 2 Report (New Reviewer)

The authors propose a mechanism related to ovarian ageing based on changes in ferritin and transferrin levels.

The paper is clearly written and requires minor language corrections. 

The manuscript is interesting and contains new findings, but has individual imperfections:

The information on brain ageing (line 50-52) is unnecessary and irrelevant to the study.

Please correct the font size in lines 319-320.

I recommend approval of the submitted manuscript with only minor amendments.

Author Response

Response to reviewer 2

Comments and Suggestions for Authors

Q1. The authors propose a mechanism related to ovarian ageing based on changes in ferritin and transferrin levels. The paper is clearly written and requires minor language corrections. 

Authors’ response: Many thanks for your kind comments. Language polishing of the manuscript has been performed by a professor in the Dept of English 

Q2. The manuscript is interesting and contains new findings, but has individual imperfections:

Authors’ response: Thanks a million for your nice remarks. Improvements have been made wherever possible.

Q3. The information on brain ageing (line 50-52) is unnecessary and irrelevant to the study.

Authors’ response: This has been done as instructed. Reference #4 about brain aging in the original manuscript has been replaced by another reference on the gonads.

Q4. Please correct the font size in lines 319-320.

Authors’ response: This has been done as instructed.

Q5. I recommend approval of the submitted manuscript with only minor amendments.

Authors’ response: Many thanks for your kind comments.

Round 2

Reviewer 1 Report (Previous Reviewer 2)

The answers to the points raised by this reviewer are only partly satisfactory.

- the authors did not comment on why they analysed IRP2 but not IRP1, which may be even more important for iron regulation than IRP2. 

- The authors stated that the discussion was reduced from 400 lines to 260 lines, which is not true, since in the pdf format the lines are now 346 vs 408 of the previous version. Halve it and focus on the original data of the work.

- a typo at line 138

Author Response

- the authors did not comment on why they analysed IRP2 but not IRP1, which may be even more important for iron regulation than IRP2.

Our response:

Thanks for reviewer’s comments.

The data on IRP1 and IRP2 are shown in Figure 2 F, H, & I and also in lines 183-185/ 339-361 /531-538 (copied below for easy reference)

  • lines 183-185:

…and the oxidation markers total nitrate/nitrite, 3-NT and HNE as well as IRP2 were upregulated in ovaries of older rats e.g.12- and 16-month- and 22-month-old compared with those of younger e.g. 3- and 9-month-old rats. IRP1 did not show striking changes (Figures 3E, F, G, H, I and J).

  • lines 339-361:

We found upregulation of proteins associated with Fe metabolism and aging ovaries involved the IRE / IRP regulatory circuit, and upregulation of ovarian TfR1 via upregulated IRP2 in naturally aging ovaries in vivo. TfR1 transports Fe into cells by binding to Tf and Ft. The post-transcriptional IRE/IRP regulatory circuit plays an essential role in Fe homeostasis in various cell types.  IRP1 and IRP2 binding to IREs in the 5’-UTR of FTH1 and Ftl mRNA inhibit their translation, whereas IRP binding to the 3’-UTR of TfR1 mRNA prevents its degradation, which then promotes TfR translation [32] and increases TfR1 expression. IRP1 expression and IRP2 expression exhibit different tissue specificities [12,33]. When there is more IRP2 in cells, IRE-dependent repression mainly inhibits the translation of Ft rather than other mRNAs, which can explain tissue-specific Fe regulation by Ft mRNAs [ 12,33]. This is the first report that upregulated IRP2 promotes TFR1 protein expression, but does not inhibit protein expression of FTH1 and Ftl chains in the naturally aging ovary (Figure 2). The upregulated FTH chain may be regulated by upregulated NF-κB at the transcriptional level (Figures 2 and 5). Fe2+ binds to 5’-UTR IREs’ stem-loop changing its conformation, which reduces its affinity for IRPs but enhances its affinity for the ribosome recruitment factor eIF4F, promoting FTH chain translation [ 34 ]. This implies that age-associated ovarian Fe2+ accumulation (Figure 3D) may contribute to the aberrantly enhanced translation of Ft via 5’-UTR IREs, and thus the aberrantly upregulated Ft in ovaries of aging rats. As aging proceeded, the ovarian iron level as well as IRP2 protein expression level increased (Figures 2F and 3I), in discord with the observation [ 35] that  an Fe-S cluster within FBXL5 (F box protein)  enhances IRP2 polyubiquitination and degradation in response to both Fe and OS. Nevertheless, the outcome of relationship between IRP2 and intracellular Fe level may differ between different tissues [ 36] . Irp2-/- mice exhibited Fe overload in the liver and duodenum but Fe deficiency in the bone marrow and spleen. Conditional Irp2 deletion in mouse liver and duodenal cells repeats the Fe overload phenotype of the liver and duodenum, indicating that Fe overload is attributed to cell-autonomous functions of Irp2 deficiency in these cells. Conditional  Irp2 deletion in splenic macrophages did not produce the splenic Fe deficiency seen in Irp2-/- mice, indicating Fe deficiency in Irp2-/- mouse spleen is secondary to Fe dysregulation in other cell types.  

  • lines 531-538:

Meyron-Holtz et al.(101) compared the consequences of genetic ablation of IRP1 to that of IRP2 in mice. Dysregulation of iron metabolism is discernible in in all tissues in IRP2-/- mice but takes place only in brown fat and the kidneys in IRP1-/- mice. IRP2 exhibits sensitivity to the iron status and can make up for the loss of IRP1 by elevating its binding activity. On the other hand, the small RNA-binding fraction of IRP1, which lacks sensitivity to cellular iron status, plays a role in basal mammalian iron homeostasis. Hence, IRP2 has central importance in post-transcriptional regulation of iron metabolism in mammals. Schalinske et al.(102) noted that the Ba/F3 family of murine pro-B lymphocytes IRP1 is not crucial for regulation of ferritin or TfR expression by iron and that IRP2 can act as the sole IRE-dependent mediator of cellular iron homeostasis. Thus IRP1 and IRP2 play different roles in regulation of iron metabolism. In our study IRP2 played a more pronounced role than IRP1(Fg 2F.2H and 2I).

- The authors stated that the discussion was reduced from 400 lines to 260 lines, which is not true, since in the pdf format the lines are now 346 vs 408 of the previous version. Halve it and focus on the original data of the work.

- a typo at line 138

Our response: The discussion has been shortened to 233 lines and the number of references has been reduced from 162 to 109. A typo at line 138 has been corrected accordingly. thanks for reviewer's comments and support.

This manuscript is a resubmission of an earlier submission. The following is a list of the peer review reports and author responses from that submission.

Round 1

Reviewer 1 Report

Title:

In my opinion, the title is too long, it should indicate the most important result encouraging reader to grab the publication.

Abstract:

  1. In general, Abstract is quite convoluted and built of too long sentences that prevent the reader from what the Authors wanted to convey.
  2. Sentence line 39-42 too long and not informative.
  3. Sentence line 42-47 too long and not informative.
  4. Term iron regulatory proteins reserved in the “bioiron society” for IRPs.

Introduction:

In the original work presented to me by Sze et al., focus on a very important aspect of ovarian physiology, natural aging. As it can be seen from reading the publication, the main role in this process is played, as it is stated by Authors by the two key proteins, which are regulated “by negative feetback”.

Specifically, Authors aimed:

  • to investigate whether two molecules involved in iron regulation by negative feedback, transferrin (via transferrin receptor) and ferritin, are aberrantly upregulated as aging progresses;

Please explain in the introduction what is negative feedback in iron metabolism. Taking into account the TFR1 mRNA regulation by the IRP / IRE system.

  • to examine the aging associated changes in iron levels and oxidative stress;
  • to verify if estradiol levels are depressed in ferritin-treated primary rat ovarian granulosa cells via the NF-κB-induced iNOS pathways;
  • to verify that NF-κB-induced oxidative stress and inflammation increases with aging and further confirm these findings by using a single intraovarian injection of an adenovirus expressing ferritin and transferrin in young rats, respectively.

Comments and Questions:

  1. Authors claim that the female ovaries age much faster than extragonadal organs, as well as the testes. Please provide reference which compare testes and ovaries in this context and showing similar aging.
  2. There is no need for a comma in verse 99.

Results:

chapter 1.

  1. Probably the title of the first chapter should be in italics.
  2. All protein abbreviations should be in parentheses.
  3. Supplementary Table 1, 2 and 3 not available.
  4. Line 111, additional space.
  5. Line 113, additional space.
  6. Line 122, additional space.
  7. On what basis the results obtained from the PDQuest and Blast2Go analysis give so categorical right to say: “Additionally, GO annotation revealed that Tf upregulated I-κB kinase/NF-κB signaling and also regulated tumor necrosis factor production and Phb protein response to nitric oxide (Supplementary Table 1). These results imply that aberrant upregulated Fth1, Ftl1, and Tf induce dysregulated iron metabolism, and upregulated I-κB kinase/NF-κB signaling mediates dysregulated metabolic processing of nitrogen compounds”?
  8. Can the proteins annotation size in Figure 1 be improved?
  9. Ftl1 is not marked in Figure 1 or is unreadable to the reader, however, as important for results, it is shown in bold in Table 1.
  10. Sentence Line 146-149 is not correct in terms of reference [12 and 13]. Intracellular iron homeostasis is mainly coordinated by the regulation of TfR1 mRNA stabilization and ferritins mRNA translation (posttranslational regulation) by IRP/IRE system. Transferrin in this case has nothing to do. Again the term, iron regulatory proteins is reserved for IRPs.
  11. Sentence 174-175, where did such conclusion come from? The increase in the level of IRP2 while the level of IRP1 does not increase does not necessarily correspond to the regulation of TfR1 mRNA by IRP2. In this case, the transregulatory activity of IRP1 is very important. The present study shows an increase in the level of nitrosative stress to which IRP1 is very sensitive due to its iron-sulfur claster (PMID: 21566147; PMID: 12746546). Thus, it is the IRP1 binding activity and not the increase in IRP2 level that may be associated with the increase in TfR1 mRNA. In this context, what is the level of TfR1 mRNA and the binding activity of IRP1?
  12. Moreover, how do the authors explain the increase in IRP2 levels as intracellular iron levels increase? Regulation of IRP2 levels is mediated by protein degradation by iron and, to a lesser extent, by oxidative stress (PMID: 32126207).
  13. Figure 2. Please show western blot analysis with at least three samples per group. Please include three samples per group and molecular weight markers (MWM) for all western blots. Please improve the quality of your WB analyzes taking into account a broader view of the blots.

chapter 2.

  1. Single words appeared in the manuscript in a much larger type.
  2. Line 205. The labile iron expression is limited to the labile iron pool (LIP), which can be both Fe3 + and Fe2 +. Same for Materials and Methods chapter.
  3. Line 208. Total iron is limited for total iron including heme and non heme iron. Same for Materials and Methods chapter.
  4. Figure 3F. Please provide NOS2 WB analysis or mRNA level.

chapter 3 and 4.

  1. Figure 4B and 5A/I. Please improve the quality of WB analyzes. They are illegible, I wonder how Authors was able to quantify the bands?
  2. Figure 4B and 5A. Is the phenomenon of the increase in the level of transcription factors related to transcription or maybe to protein stabilization?

chapter 5 and 6.

  1. Please provide the original GSTP1 and SOD2 blot scans for Figure 6.
  2. What is the protein level for TfR1 in Figure 7B? Such information will enable the correct drawing of conclusions.
  3. Change Flc and Fhc to Ftlc (Ftl1) and Fthc (Fth1) in Figure 7.

Discussion and conclusions:

Due to my numerous comments on the quality of the presented results, I am not able to address many of the issues included in the discussion. Undoubtedly, the work requires major revision and intensive linguistic revision.

Methodology:

There are methodological deficits related to the poor quality of the WB blots. It would also be advisable to analyze the qPCR in several parts of the work. The job seems incomplete.

Best regards.

Author Response

Please find attached reply, thank you.

Reviewer 2 Report

The work analyses the ovaries of young and aged rats and in a preliminary proteomic study finds that ferritins and transferrin are higher in the aged rats. This was confirmed by western blots. Further analyses showed that estradiol decreased with aging, while iron and indices of oxidative stress increased. This was accompanied by upregulation of NF-kB with consequent deregulation of iNOS. The effects of ferritins were studied also by adding them to primary granulosa cells and by inducing their expression in the ovaries by adenovirus injections. The work is original and the data are interesting, but the text seems too long and difficult to read. Some experiments are not convincing.

- The title is much too long and not really encouraging the reader. Similar for the abstract, that should just summarize the experiments more clearly. In addition, "iron overload" is not really proper for the minor iron increase, and "iron regulatory proteins" do not apply to ferritins + transferrin.

- The discussion is 5 pages long and it should be strongly reduced, reorganized, and more focused.

- The in vitro experiments of fig 4 should be more detailed. Which type of ferritin was used, did it contain iron, was it taken up by the cells?

- Also the experiments in fig 7 are not fully convincing. The H ferritin increase by adenovirus is very minor, and it is surprising that it has the same effect on the ovaries as L ferritin and transferrin. In materials and methods, the control is the empty vector, while in the text it is GFP, which is not described.

- The authors may discuss why they incubated the primary granulosa cells with ferritin instead of iron that would stimulate ferritin expression and cause iron load. Also, the use of iron chelators would be informative.

Author Response

Please find attached reply, thank you.

Round 2

Reviewer 1 Report

Dear Authors

Thank you for making changes. However, the Authors did not understand my intentions regarding the introduction to the introduction chapter of the regulation of the stabilization of the 3 'UTR mRNA of TfR1  and the inhibition of translation at the 5' UTR mRNA of ferritin mRNA by the IRP/IRE system. One cannot talk about the regulation of TfR1 and ferritin without considering this system and this regulation. By no means did I mean Hepcidin regulating Fpn. Anyway, this introduced fragment (in green) is very badly written, there are numerous errors, no references to literature, bad English.

In general it cannot be explained by the inability to improve the publication quality by the fact that one of the authors has disappeared and has no contact with him.  Why the authors did not bother to get the level of TfR1 mRNA?

I regret and certainly support my negative opinion of the publication.

Reviewer 2 Report

The answers to the points raised by the reviewer are rather satisfactory, and the manuscript improved somehow. However, some minor problems remain.

- The statement that iron-containing ferritin was used with the granulosa cells is not sufficient. This is an important point of the work that is expected to clarify if ferritin protein or its iron affects the cells. Thus, the amount of ferritin iron should be stated, and also the type of ferritin: human or murine, recombinant or natural, H or L chain?. The ferritin uptake by the cells should be verified by WB.

- In the abstract, L. 38 “adenovirus expressing iron regulatory proteins” is not correct, the system expressed ferritins and transferrin.

- L. 58 “Aging-associated systemic iron accumulation due to an elevated serum level of ferritin” serum ferritin level does not cause iron accumulation, it is a marker of iron accumulation.

- L. 158 “Thus, the IRP1 binding activity and not the increase in IRP2 level may be associated with the increase in TfR1 mRNA.” The sentence is unclear since IRP1 activity was not measured. Remove or change it. 

- L.208 “Ovarian granulosa cells can take up ferritin[32].» the reference is about cationic ferritin, and does not apply to normal ferritins.

- A comment should be made about the very low efficiency of adeno-FTH vector, and why nevertheless it produced strong effects on NF-kB and other indices.